# Dream to Manipulate: Compositional World Models Empowering Robot Imitation Learning with Imagination

**Leonardo Barcellona**[1,2*]   **Andrii Zadaianchuk**[3]   **Davide Allegro**[1]   **Samuele Papa**[3]
**Stefano Ghidoni**[1]   **Efstratios Gavves**[3,4]

[1]University of Padua, Italy   [2]Politecnico di Torino, Italy
[3]University of Amsterdam, The Netherlands   [4]Archimedes/Athena RC, Greece

## Abstract

A world model provides an agent with a representation of its environment, enabling it to predict the causal consequences of its actions. Current world models typically cannot directly and explicitly imitate the actual environment in front of a robot, often resulting in unrealistic behaviors and hallucinations that make them unsuitable for real-world robotics applications. To overcome those challenges, we propose to rethink robot world models as *learnable digital twins*. We introduce DreMa, a new approach for constructing digital twins automatically using learned explicit representations of the real world and its dynamics, bridging the gap between traditional digital twins and world models. DreMa replicates the observed world and its structure by integrating Gaussian Splatting and physics simulators, allowing robots to imagine novel configurations of objects and to predict the future consequences of robot actions thanks to its compositionality. We leverage this capability to generate new data for imitation learning by applying equivariant transformations to a small set of demonstrations. Our evaluations across various settings demonstrate significant improvements in accuracy and robustness by incrementing actions and object distributions, reducing the data needed to learn a policy and improving the generalization of the agents. As a highlight, we show that *a real Franka Emika Panda robot*, powered by DreMa's imagination, can successfully *learn* novel physical tasks from just *a single example* per task variation (one-shot policy learning). Our project page can be found in: https://dreamtomanipulate.github.io/.

## 1 Introduction

World models are learnable representations of the real world that an agent can use to predict the consequences of its actions (Ha & Schmidhuber, 2018). They predict future states given the current state and an action, empowering the agents to learn new skills from the inferred evolution of the model (Hafner et al., 2019; 2020; Gumbsch et al., 2024; Hansen et al., 2024; Yang et al., 2023). Thus, they are a fundamental building block for robots to imagine, a capability that we humans exploit to interact with new environments (Hawkins et al., 2017; Ha & Schmidhuber, 2018).

Applying world models to real robots implies rendering realistic observations (Yang et al., 2023; Zhou et al., 2024) while simulating the dynamics (Li et al., 2024). Whereas most of the world models learn a statistical representation of the dynamics (Hafner et al., 2019; Gumbsch et al., 2024; Hansen et al., 2024; Yang et al., 2023), it is unrealistic to expect any training distribution to encompass all possible configurations of the world. This limits the current literature, as agents need the ability to imagine beyond direct experience. We identify three requirements needed by an effective world model for a robotic agent. First, it must be inherently *compositional*, allowing the generation of novel and valid combinations of learned concepts (Zhou et al., 2024). Second, it must be *object-centric*, as the world is primarily composed of objects of varying number, type, and

---

*Correspondence to: leonardo.barcellona@polito.it, {a.zadaianchuk, e.gavves}@uva.nl

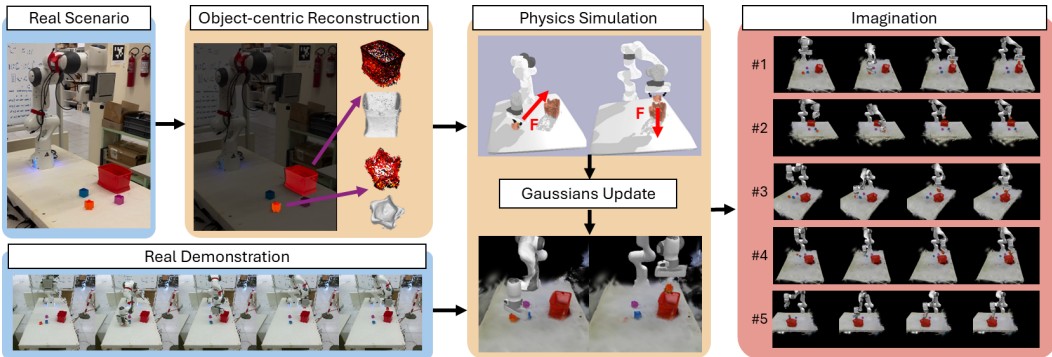

Figure 1: Overview of imagination with DREMA, which builds a compositional manipulation world model from environment images using object-centric Gaussian Splatting to generate novel demonstrations by transforming real ones.

configuration (Spelke, 1990; Kipf et al., 2019). Third, *manipulating* the model should be simple to ensure that actions and consequences can be controllably modelled (Li et al., 2024).

In this work, we propose a *compositional manipulation world model*, named DREMA, that allows robotic agents to exploit physically reliable predictions to generalize to novel situations. Thus, it is an interactable physics-constrained representation that bridges robot imagination with real-world demonstrations following recent advances in learnable digital twins and explicit world models, which both enable agents to simulate and reason about their environment (discussed in App. E). We shape DREMA by reconstructing the scene with object-centric Gaussian Splatting (Kerbl et al., 2023) and by embedding the obtained compositional representation into a physics simulator, e.g., PyBullet (Coumans & Bai, 2016). Gaussian Splatting reconstructs 3D scenes as a set of independent Gaussians that can be optimized and rendered, allowing for extremely high resolution and real-time graphics rendering. We infer compositional properties of rendered Gaussian Splats using foundational models (Ravi et al., 2024; Ren et al., 2024), allowing the agent to manipulate the object-centric model to create new scenes.

Through DREMA, it is possible to execute robot policies directly in imagination while rendering realistic observations. This capability benefits imitation learning, whose objective is to solve tasks given example demonstrations (James et al., 2022). To ensure a reliable generalization of the learned policy, demonstrations should contain sufficient variations, which is impractical and time-consuming with real applications (Mandlekar et al., 2023). We show that imposing equivariant transformations, namely roto-translation of the demonstrated actions and the objects, DREMA produces new high-quality data inside the world model and reduces the required demonstrations. We exploit imagination to generate new demonstrations that augment the original ones, while also using the proposed world model to verify that the new actions result in successful task execution and rendering of realistic observations.

Our contributions are: (1) We are the first to propose a compositional manipulation world model, DREMA, that empowers robots with the ability to imagine in a grounded manner; (2) We propose to exploit the compositional structure of the world model to generate novel, valid, and realistic training demonstrations for imitation learning; (3) We empirically demonstrate in simulations and with real robots that DREMA improves agent generalizability even with a minimal number of demonstrations.

## 2 RELATED WORK

### 2.1 WORLD MODELS

World models were originally conceived to improve policy learning in reinforcement learning by predicting the consequences of actions on the environment (Ha & Schmidhuber, 2018; Hafner et al., 2019; 2020; 2023). State-of-the-art approaches often leverage generative models to build these world models (Jia et al., 2023; Hu et al., 2023; Yang et al., 2023; Bruce et al., 2024). While initially applied to game simulations (Ha & Schmidhuber, 2018; Hafner et al., 2019) and navigation tasks (Hu et al., 2023; Jia et al., 2023), world models are now being extended to robotics (Yang et al., 2023; Wu

et al., 2023; Zhou et al., 2024). For example, UniSim (Yang et al., 2023) uses diffusion to generate possible future frames conditioned on robot actions. DayDreamer (Wu et al., 2023) demonstrates that skills such as locomotion and manipulation can be learned using Dreamer (Hafner et al., 2019), while RoboDreamer (Zhou et al., 2024) uses compositionality to generate novel actions. The main limitation of these approaches is their inability to comprehensively capture the motion dynamics and physics specific to the robot's environment, particularly when faced with scenarios outside their training domain (Yang et al., 2023). By contrast, we propose a compositional manipulation world model that integrates cutting-edge photorealism with Gaussian Splats and realistic dynamics with physics engines, showing we can tailor them for efficient imitation learning in real-world scenarios.

## 2.2 SIMULATION-BASED RECONSTRUCTIONS

Thanks to recent advances in reconstruction (Kerbl et al., 2023) and in physics simulations (Coumans & Bai, 2016; Li et al., 2019), in the last year many have proposed to combine both for realistic simulations of actual environments (Xie et al., 2024; Jiang et al., 2024; Meyer et al., 2024; Lou et al., 2024; Abou-Chakra et al., 2024; Ruan et al., 2024; Torne et al., 2024). Particle-based simulators are capable of modeling more complex dynamics than traditional rigid body simulators (Li et al., 2019), making them beneficial for robotics (Chen et al., 2024a). Recently, neural radiance fields have been applied to implicitly learn dynamic systems (Li et al., 2023; Whitney et al., 2024). The introduction of Gaussian Splatting (Kerbl et al., 2023) has opened avenues for novel approaches that bridge these simulation techniques with high-quality rendering methods (Xie et al., 2024; Jiang et al., 2024). However, they are not compositional (Xie et al., 2024) or object-centric (Xie et al., 2024; Jiang et al., 2024). PEGASUS (Meyer et al., 2024) combines Gaussian representations with mesh simulations to improve the 6D pose estimation, while Abou-Chakra et al. (2024) and Lou et al. (2024) show that Gaussian representations can be adapted to track moving objects. Both approaches do not allow agents to predict future states, rather focusing more on tracking in observed past trajectories. In contrast, we empirically demonstrate that our approach predicts useful estimation of future states. While simulating a replica of the environment can benefit policy evaluation (Li et al., 2024), two recent studies showed that scene reconstruction can enhance robot policies (Ruan et al., 2024; Torne et al., 2024). Ruan et al. (2024) reconstruct the environment to deform robot trajectories, while our object-centric approach focuses on the interaction between objects and the agent. Although being close to our approach, Torne et al. (2024) requires to *manually model* the object-centric representation and needs adaptation between the simulation and the real world. Instead, we leverage the zero-shot capabilities of foundational models (Kirillov et al., 2023; Ren et al., 2024; Cheng et al., 2023) to automatically create a compositional world model and show the effectiveness of using transformed demonstrations for task learning without requiring extensive interaction with the world model.

## 2.3 IMITATION LEARNING AND DATA AUGMENTATION

Imitation learning involves acquiring task knowledge from a dataset of demonstrations (James et al., 2022). Vision-based imitation learning focuses on deriving policies directly from RGB or RGB-D images (Zeng et al., 2021; Shridhar et al., 2023; 2022; Goyal et al., 2024). PerAct (Shridhar et al., 2023), for instance, voxelizes the environment to infer actions in 3D. However, despite advancements in generalization, many approaches (Ze et al., 2023; Goyal et al., 2024) still require hundreds of demonstrations to learn robust policies. MIRA (Lin et al., 2023) employs NeRF's novel view synthesis to simulate new camera views of the environment, while Lu et al. (2025) applied Gaussian Splatting for action inference. In contrast, our approach leverages Gaussian Splatting to generate not only novel views but also new environmental configurations, enhancing generalization.

Data augmentation is an established technique (Krizhevsky et al., 2012; He et al., 2016). However, its application in robotics is challenging, as altering robot actions can unpredictably affect environments and tasks (Pitis et al., 2022). Despite this, Laskin et al. (2020) demonstrated that even simple augmentations can improve robot policies. Approaches such as generating counterfactual transitions (Pitis et al., 2022) and applying invariant transformations (Corrado & Hanna, 2024) enhance learning by expanding training datasets. However, these methods primarily recombine existing trajectories rather than generate entirely new configurations. In contrast, our approach imagines novel equivariant configurations of objects and actions, enabling more diverse and effective data augmentation. A related method, MimicGen (Mandlekar et al., 2023), uses object-centric

Figure 2: Steps to create the compositional world model with DREMA: observation of the environment and scene decomposition, representation extraction and future predictions.

segments for compositional data generation but requires multiple demonstrations. Our method achieves comparable benefits while working effectively with as few as one demonstration. Whereas some authors argue that imagining various object configurations is unrealistic (Zeng et al., 2021), we demonstrate that this is achievable in real-world scenarios.

# 3 DREAM-TO-MANIPULATE WITH GAUSSIAN SPLATS & PHYSICS ENGINES

We want to construct a *compositional manipulation world model* $\mathcal{M}$, that is (i) explicitly compositional, (ii) object-centric, and (iii) controllable, allowing the agent to control all the objects inside $\mathcal{M}$. The model $\mathcal{M} = \left( \mathcal{O}_t, \mathcal{A}_t, D, \mathcal{T} \right)$ comprises a set of $K$ object assets in 3D, $\mathcal{O}_t = \{o_{k,t}\}_{k=1}^K$ at time $t$; the model of the agent $\mathcal{A}_t$ that manipulates the objects in the world; the dynamics $D$ of the world in the form of an operator function, $\left( \mathcal{O}_{t+1}, \mathcal{A}_{t+1} \right) = D \cdot \left( \mathcal{O}_t, \mathcal{A}_t \right)$, which —given the actions of the agent $\mathcal{A}$ at time $t$— transforms the states of object assets $\mathcal{O}_t$ and agent's state $\mathcal{A}_t$; the manipulation tasks $\mathcal{T}$ expected from the agent. For clarity, we drop the time index $t$ from $o_{k,t}$ when this can be inferred from the context. The input to the algorithm that constructs the compositional manipulation world model is simply an RGB-D video $\mathbf{X} = (x_1, ..., x_N)$ of $N$ frames.

In the following, we first discuss our compositional 3D scene representation based on Gaussian Splats, which forms the basis for our world model. Afterward, we detail how we obtain each component of the world model given our 3D scene representation.

## 3.1 REPRESENTING 3D OBJECTS WITH GAUSSIAN SPLATTING

Central to the compositional manipulation world model is the inverse graphics function $\mathcal{G} = h(\mathbf{X})$ using Gaussian Splatting (Kerbl et al., 2023). The inverse graphics function returns a 3D representation of the world $\mathcal{G}$ given a sequence of $n$ images $\mathbf{X} = (x_1, ..., x_n)$, collected as a video (Kerbl et al., 2023). The key idea behind Gaussian Splatting is to ground many Gaussian blobs $g = (p, r, s, \alpha, c), \forall g \in \mathcal{G}$ physically in the 3D space. Per Gaussian, we have $p = [p_x, p_y, p_z]$ as its center , its orientation $r$ and scale $s = (s_u, s_v)$ (the covariance), the encoded color $c$, and the opacity $\alpha$ of the color. That is, given any arbitrary 3D location and angle in space (corresponding to a 2D camera plane), Gaussian Splats can render a projected 2D image that resembles the true observed object (or scene). For rendering the appearance onto the 2D image, one simply $\alpha$-blends the colors of the Gaussians along each of the rays that connect the 3D point to the points on the 2D projected image. In the end, the Gaussian Splatting algorithm optimizes the Gaussian parameters in $g$ to render the objects and the scene photorealistically for both known and unknown 3D locations (named novel view synthesis). Gaussian Splats have taken the graphics community by storm (Huang et al., 2024; Xie et al., 2024; Meyer et al., 2024) not only for their exceptional photorealism but also for their real-time computation. For further details, we refer to the seminal work of Kerbl et al. (2023).

**Object-centric Gaussian Splats.** Gaussian Splats generate photorealistic renderings of the 3D scene that are unstructured in terms of object semantics. Instead, we are interested in object-centric representations. To obtain object-centric Gaussian Splats, together with the input 2D images in $\mathbf{X}$ for our $K$ objects in our world, we introduce sets of segmentation masks, $\mathbf{Y}_k \in \mathcal{Y}, k = 1, \ldots, K$. We consider $k$ to be varying so that not constrain in advance the number of objects in our world. Each object-set of segmentations $\mathbf{Y}_k = \left( y_{1,k}, ..., y_{n,k} \right)$ contains the 2D masks for the $k$-th object in all $N$ images in $\mathbf{X}$. With segmentation masks $\mathbf{Y}_k = \left( y_{1,k}, ..., y_{n,k} \right)$ for each object $k$ and for all $n$ frames,

we can simply mask the respective video frames, $y_{n,k} \odot x_{n,k}$, to obtain the object-centric instances, which we can then feed to the Gaussian Splatting. Since in our setting the robot is also equipped with depth sensors, we further strengthen the original Gaussian Splatting optimization by including the depth measurements $x^{\text{depth}} \in \mathbf{X}_{\text{depth}}$ for additional supervision, $\mathcal{L} = \mathcal{L}_{\text{rec}} + \lambda_{\text{n}} \mathcal{L}_{\text{n}} + \lambda_{\text{depth}} \mathcal{L}_{\text{depth}}$. Here $\mathcal{L}_{\text{rec}}$ and $\mathcal{L}_n$ are the reconstruction and normal losses defined in the original 2DGS. The extra depth regularization, $\mathcal{L}_{\text{depth}} = \|\hat{x}_{GS}^{\text{depth}} - x^{\text{depth}}\|_1$, uses the $\ell_1$ norm to penalize Gaussians that have outlier depth. We set the hyperparameters $\lambda_{\text{normal}}$ and $\lambda_{\text{depth}}$ for all experiments empirically as in Huang et al. (2024). Our object-centric inverse graphics function takes the form $\mathcal{G}_k = h(\mathbf{X}, \mathbf{X}_{\text{depth}}, \mathbf{Y}_k)$.

**Deformable and articulated objects.** It is important to note that since the standard Gaussian Splats only work with (static) scenes, we cannot account for deformable or articulated objects. An exciting future direction is to bootstrap on the significant progress in Dynamic Gaussian Splatting (Luiten et al., 2023) to automatically model also articulation (see App. G) and deformation.

Next, we describe how to decompose the scene into $k$ sets of regions defined by segmentation masks $\mathcal{Y}$ to obtain object assets and model dynamics.

### 3.2 Decomposing the Scene into Object-centric Regions

Decomposing the high-dimensional scene representation into objects (Locatello et al., 2020; Seitzer et al., 2023; Wu et al., 2024) that are independently controllable greatly simplifies control and the manipulation of the dynamics in the environment (Zadaianchuk et al., 2021; 2022). Inspired by recent work on embeddable 3D assets (Deitke et al., 2023; Wu et al., 2024), we ideally want to learn Gaussian Splats that belong to the same object. We take advantage of the zero-shot capabilities of open-vocabulary tracking models (Cheng et al., 2023; Wang et al., 2023; Ravi et al., 2024) to ensure that we obtain consistent predictions. Our compositional world model uses the DEVA open-vocabulary tracker to extract the objects and their segmentation masks $\mathcal{Y}_k$ in the video $\mathcal{X}$ of the scene. Since DEVA discovers objects given prompts, we use general and minimal prompts: `object` for all the objects and `table` for segmenting the table of our table-top robot. Having computed the segmentation masks $\mathcal{Y}_k = (y_{1,k}, ..., y_{n,k})$ object $k$ and for all $n$ frames, we can learn object-centric Gaussian Splats as described in Sec. 3.1. Advances in the object-centric (Didolkar et al., 2024; Zadaianchuk et al., 2024; Wu et al., 2024) and open-vocabulary (Ravi et al., 2024; Bianchi et al., 2024) literature can be seamlessly integrated into our compositional manipulation world model.

### 3.3 Modelling Robot-Object Dynamics with Object Assets

To be able to predict the consequences of the actions of the robot, we must model the dynamics $D$. We can directly use external physics engines (Todorov et al., 2012; Coumans & Bai, 2016) for the dynamic operator $D$. For our model, we rely on `PyBullet` (Coumans & Bai, 2016) for its simplicity.

While object-centric Gaussian Splat representations $\mathcal{G}_k$ are good for photorealistic renderings of objects, they are not designed for physics simulations. Instead, physics engines —including `PyBullet`— typically operate on object meshes. We thus need to convert $\mathcal{G}_k$ into object mesh grids, $M_k = (\mu_{k,\cdot}, \rho_{k,\cdot}, u_{k,\cdot})$, with mesh centers $\mu_{k,\cdot}$, orientations $\rho_{k,\cdot}$, and appearances $u_{k,\cdot}$, to obtain our world object assets, $o_k = (\mathcal{G}_k, M_k, \omega)$. With $\omega$ we explicitly denote the relevant physical parameters of the world, such as the masses of objects or the friction coefficient. In the robot simulator, these are in the URDF files (Villasevil et al., 2024). During learning the policies, the parameters $\omega$ can be either given or inferred (Memmel et al., 2024). The physics engine acts as an operator on the positional and orientation parameters of the mesh grid of all objects in the world, that is

$$\begin{pmatrix} \mu_{\cdot,t+1} \\ \rho_{\cdot,t+1} \end{pmatrix} = \begin{pmatrix} \mu_{\cdot,t} + \Delta\mu_{1...K} \\ \Delta\rho_{1...K} \cdot \rho_{\cdot,t} \end{pmatrix}, \tag{1}$$

using standard Newtonian mechanics as the dynamics operator. Note that any object in the world, including the robot arm, can interact with any other object. This is why in Equation 1 we drop the object subscript $k$, and why in the differential $\Delta$ we add all objects $1 \ldots K$. Knowing the forces that the robot exerts on the environment at any moment, our compositional manipulation world model can easily *imagine* the future states of the world and the consequences of the robot's actions. Finally, we apply the change in mesh center $\Delta\mu_k$ and mesh orientation $\Delta\rho^k$ to all Gaussians $g \in \mathcal{G}_k$ by updating their centers $p' = p + \Delta\mu_k$ and their orientations $r' = \Delta\rho_k \cdot r$.

**Modelling the agent.** In the above, we focused on the objects or our world. The robot is also an object that is self-visible, thus it can be just as easily represented as an object-centric mesh grid from the corresponding Gaussian Splat reconstruction. A difference of the robot arm compared to other objects is that it is articulated. However, the robot joints are known in advance, and we can use their associated segmentation masks to link the different robot parts, ensuring accurate Gaussian representation for rendering purposes. Regarding the dynamics of the robot arm, our assumption is that the robot can model its own dynamics perfectly, similarly as humans have an accurate internal model (Wolpert et al., 1998). This is a reasonable assumption, since most manipulation robots have the corresponding URDF file, allowing for the instantiation of a simulator with the robot model.

**World model verification from demonstrations.** So far we assumed that the physical parameters are fixed. With this assumption, we cannot guarantee that the manipulation world model will lead to correct predictions for arbitrary actions, as errors in the Gaussian Splat reconstructions and their mesh grids, or misalignment in the physical parameters, could result in cumulative errors when applying the dynamics. That said, our undertaking in this work is that by adopting explicitly grounded appearance and dynamics models for the world and its objects, we can still obtain a useful model. Suppose that the model predictions for replaying the sequence of actions in the demonstration result in a final state similar to the final state of the demonstration. In that case, we can still use the world model as an interactive and compositional representation of the demonstration itself. In the next section, we show that such an interactive representation allows us to augment the original demonstrations with imagined ones to train a more robust imitation learning policy.

## 3.4 ENGINEERING

The proposed compositional manipulation world model is a significant undertaking, requiring extensive engineering to ultimately be effective with real-world robots. However, one of the key advantages of this approach is that it enables *a real robot* to learn novel policies using only a single example. We describe the most critical engineering components in detail, with additional information provided in App. J. While Gaussian Splatting is primarily designed for novel view synthesis, we observed unexpected behavior at low resolutions, leading to incorrect action predictions by the robot. Additionally, depth information rendered was inaccurate, as the original Gaussian Splatting method does not ensure that Gaussians align with surfaces. To address this, we implemented 2DGS Gaussian Splatting (Huang et al., 2024). For successful real-world deployment of DREMA, properly aligning the reconstruction and the simulator is crucial, which we achieved using a calibrated camera mounted on the robot (Allegro et al., 2024). Also, estimating physical parameters is a complex field in its own right (Mehra, 1974; Chebotar et al., 2019; Gao et al., 2022; Huang et al., 2023), and methods such as ASID (Memmel et al., 2024) or AdaptSim (Ren et al., 2023) can be applied for this purpose. In this paper, for simplicity we pick constant physical parameters that seemed to work well in the validation sets.

## 4 EQUIVARIANT TRANSFORMATIONS FOR IMITATION LEARNING

In computer vision, many data augmentations preserve the semantic information of an image, enabling robust recognition and self-supervised representation learning. However, augmenting demonstration data for imitation learning presents a greater challenge (Pitis et al., 2020; Urpí et al., 2024). When altering an observation sequence (*e.g.*, by changing an object's initial position), it is necessary to also modify the corresponding action sequence to maintain a semantically consistent demonstration—one that still solves the intended task. Therefore, generating valid augmentations in imitation learning requires *equivariant transformations* that simultaneously adjust both the high-dimensional observations and the associated actions.

In imitation learning, the agent learns from a dataset $\mathcal{Z} = \{\zeta_1, \ldots, \zeta_M\}$ consisting of $M$ demonstrations, each paired with a corresponding task encoding $\mathcal{T} = \{t_1, \ldots, t_M\}$. When tasks are described by language-based goals, the task $t_m$ is represented by a language embedding (e.g., pre-trained CLIP embeddings) corresponding to the phase that describes the task. Each demonstration $\zeta_m$ consists of a sequence of continuous actions $\mathbf{A} = (a_1, \ldots, a_t)$, which are encoded as end-effector poses and gripper states. These actions are paired with observations $\mathbf{Q} = (q_1, \ldots, q_t)$, which in our case can be the object assets from the compositional manipulation world model.

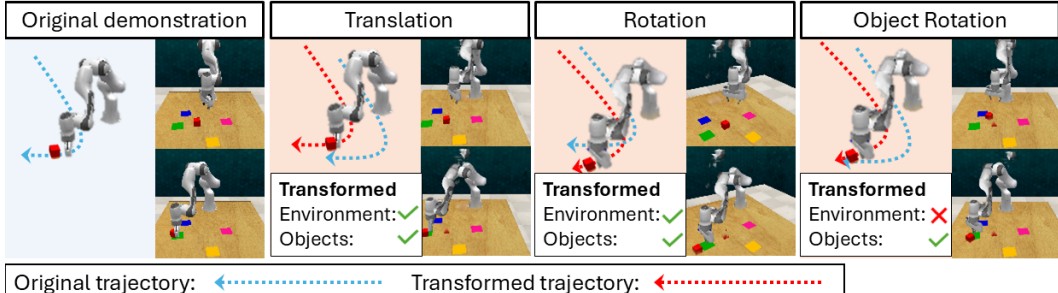

Figure 3: The effect of equivariant translation, equivariant rotation, and the object rotation transformations. Top row: start of demonstration. Bottom row: target of demonstration.

**Equivariant transformations for imagining novel demonstrations.** Finding equivariant transformations for high-dimensional observations $\mathbf{Q}$ and actions $\mathbf{A}$ is typically challenging, but DREMA simplifies this by using explicit representations for object assets $\mathcal{O}$. In our world model, both the robot's actions $\mathbf{A}$ (the 3D positions of the end-effector and gripper) and the 3D positions of the objects, including initial and target locations, are transformed equivariantly under any rigid transformation. As a result, these transformations preserve task success in the imagined demonstrations. This allows us to generate novel "imaginations" by applying geometric transformations to the objects' positions and the robot's actions, without changing the robot's internal state. By executing these transformed actions $\hat{\mathbf{A}}$ and rendering the corresponding observations $\hat{\mathbf{Q}}$ in DREMA, we generate new demonstrations $\hat{\zeta} \in \mathcal{Z}_{new}$ not present in the original dataset $\mathcal{Z}$. In the following, we describe several such transformations applicable to manipulation tasks (see Fig. 3).

**Roto-translation of the environment and objects.** Given the roto-translation $R_\kappa$ and the point of application $P_\kappa$, we transform each object asset $o_k$ at time $t = 0$ (the beginning of the demonstration) and the Gaussians of the environment. To adapt the same demonstration to the transformed environment $R_\kappa$ is applied to the end-effector translation $et_i$ and the end-effector rotation $er_i$ of each action $a_i = (et_i, er_i, ga_i)$. Consequently, the agent (fixed) is required to execute the transformed set of actions in the new environment.

**Object rotation.** This transformation involves rotating the objects around the final position of the action sequence $\mathbf{A}$. The transformation $R$ is applied to both the object poses and the action trajectory $\mathbf{A}$, keeping the demonstration consistent with the original task. In this second transformation we fixed $P_\kappa = et_t$ (the last position of the end-effector). We then rotate the objects assets $o_k$ at time $t = 0$ and the actions $a_i$ similarly to the previous approach. In this transformation we do not change the environment since the trajectory brings the object to a similar position.

**Verification of transformed demonstrations.** Although these transformations can generate novel demonstrations, there is a risk of producing invalid outcomes due to factors like the robot's inability to execute the entire sequence or inaccuracies in the world model dynamics. Therefore, it is crucial to verify that the generated demonstrations still solve the intended task. In a general setting, a success detector would be required to ensure task completion. However, for goal-based tasks where equivariant transformations are applied, it is sufficient to confirm that the final positions of the objects in the transformed trajectory remain close to their expected transformed goal positions $\hat{l}_i$: $\|\hat{l}_i - R_i(l_i - P_{\kappa,i}) + P_{\kappa,i}\| < \tau$ , where $R_i$ represents the transformation applied, $P_{\kappa,i}$ is the center of the transformation, and $\tau = 0.015$ meters is the empirical threshold for positional accuracy.

## 5 EXPERIMENTS

We evaluate the performance of DREMA in both simulated and real-world environments, comparing the effectiveness of imagination-generated data to imitation learning agents trained only on original demonstrations. For this evaluation, we use PerAct (Shridhar et al., 2023), a common baseline for imitation learning. Our study addresses the following key questions: 1. How does DREMA perform with only a few demonstrations available, in single- and multi-task settings? Does our manipulation

Table 1: Comparison of trained on original, augmented, and DREMA with imagined and all demonstrations, reporting mean ± std and max success over 5 runs on 50 test environments.

| | Single-task | | | | | | | | | |
|---|---|---|---|---|---|---|---|---|---|---|
| | *close jar* | | *insert peg* | | *lift* | | *pick cup* | | *sort shape* | |
| | mean ± std | max | mean ± std | max | mean ± std | max | mean ± std | max | mean ± std | max |
| PerAct *(Original data)* | 38.4 ± 0.80 | 40 | 0.0 ± 0.00 | 0.0 | 22.8 ± 1.6 | 26 | 13.2 ± 2.04 | 16 | 6.4 ± 1.50 | 8 |
| Random patches (Laskin et al., 2020) | 45.2 ± 3.49 | 48 | 2.0 ± 1.79 | 4 | 20.4 ± 2.65 | 24 | 37.2 ± 3.25 | 42 | 3.6 ± 2.94 | 8 |
| Random table color (Chen et al., 2023) | 45.0 ± 1.73 | 46 | 1.6 ± 1.50 | 4 | 23.6 ± 0.80 | 24 | 25.6 ± 2.94 | 30 | 8.0 ± 1.26 | 10 |
| Distractors (Bharadhwaj et al., 2024) | 36.4 ± 0.80 | 38 | 0.4 ± 0.80 | 2 | 22.8 ± 1.60 | 24 | **41.2 ± 2.99** | **44** | 4.4 ± 2.33 | 8 |
| DREMA *(imagination)* | 41.2 ± 2.40 | 46 | 1.2 ± 0.98 | 2 | 17.2 ± 1.6 | 20 | 28.0 ± 4.20 | 36 | 9.6 ± 1.50 | **12** |
| DREMA + Original *(All data)* | **51.2 ± 1.60** | **54** | **2.4 ± 2.33** | **6** | 23.6 ± 1.50 | 26 | 34.4 ± 3.88 | 40 | **11.2 ± 1.60** | **12** |
| | *place wine* | | *put in cupboard* | | *slide block* | | *stack blocks* | | *avg single-task* | |
| | mean ± std | max | mean ± std | max | mean ± std | max | mean ± std | max | mean | max |
| PerAct *(Original data)* | 10.0 ± 1.79 | 12 | 2.0 ± 0.00 | 2 | 48.4 ± 3.20 | 50 | 2.8 ± 0.98 | 4 | 16.0 | 17.6 |
| Random patches (Laskin et al., 2020) | 12.8 ± 2.40 | 16 | 2.4 ± 1.50 | 4 | 36.8 ± 0.98 | 38 | 0.0 ± 0.0 | 0.0 | 17.8 | 20.4 |
| Random table color (Chen et al., 2023) | 10.0 ± 2.83 | 14 | 1.6 ± 0.80 | 2 | 40.8 ± 4.32 | 46 | 7.2 ± 2.04 | 10 | 18.2 | 20.7 |
| Distractors (Bharadhwaj et al., 2024) | 14.8 ± 4.49 | 22 | **3.2 ± 0.98** | **4** | 49.2 ± 0.98 | 50 | 8.4 ± 2.94 | 12 | 20.1 | 22.7 |
| DREMA *(imagination)* | 16.0 ± 2.19 | 18 | 0.4 ± 0.80 | 2 | 54.4 ± 2.15 | 62 | 4.0 ± 0 | 4 | 19.1 | 22.4 |
| DREMA + Original *(All data)* | **26.8 ± 4.30** | **32** | 2.8 ± 0.98 | **4** | **62.0 ± 2.19** | **66** | **11.6 ± 1.96** | **14** | **25.1** | **28.2** |

world model still provide an advantage when more demonstrations are available? 2. How do the different components of DREMA contribute to overall performance? 3. How well does DREMA scale to real-world tasks, where both imitation learning and world model training are more complex?

## 5.1 ONE-SHOT POLICY LEARNING: SINGLE- AND MULTI-TASK

We test DREMA with a minimal number of demonstrations per task, i.e., one example per task variation up to five maximum examples (five examples for each task except for *slide block* and *place wine* that have 4 and 3 variations). We compare PerAct trained with original demonstrations, PerAct trained with DREMA's imagined ones, and PerAct trained with imagined and original combined. All simulation experiments are conducted on 50 environment configurations, with the test repeated five times to account for variability introduced by the motion planner.

**Data collection in simulation.** We replicate the experimental setup of PerAct (Shridhar et al., 2023) using RLBench (James et al., 2020), which involves a Franka Emika Panda robot with a parallel jaw gripper, placed on a table with objects that vary in color and position. We focus on tasks excluding articulated objects, as our current object-centric priors do not handle articulations. The experiments cover 1) non-prehensile tasks and 2) pick-and-place tasks (nine tasks in total). Following Ze et al. (2023), we collect a sequence, $\mathbf{X}$, of 200 images by simulating a rotating camera around the scene. PerAct supports multi-task training, where a single policy adapts to multiple tasks based on prompts. We train both PerAct and DREMA on three tasks using the same demonstrations as in the single-task setup. Hyperparameters are consistent with the original paper (Shridhar et al., 2023), except for the training iterations and batch size (see App. L). To prevent overfitting, we validate all models performance on a separate set for each 10k training iterations for multi-task and 5k for single-task and select the best-performing model.

**One-shot single- and multi-task results (Table 1 & Table A.5).** Since the limited number of demonstrations is insufficient to learn a general policy, the PerAct model trained solely on original demonstrations performs the worst. In contrast, DREMA, exploiting around 800 demonstrations (see App. L), significantly outperforms PerAct with the original data in single-task settings (Table 1), achieving an average accuracy improvement of +9.1%. This result highlights that additional imagination-based demonstrations help create more robust policies, even when the original dataset is small. When DREMA is compared to invariant augmentations —namely, random patches, random table color, and random distractors— it achieves the best performance, with a 5.0% average improvement over the best invariant augmentation (see App. A). Additionally, DREMA improves performance in all tasks compared to PerAct, differently from invariant augmentations. In multi-task settings (Table A.5), DREMA is tested on *close jar*, *slide block*, and *sort shape* simultaneously. It achieves an average accuracy of 35.5%, representing a +13.1% improvement over PerAct, demonstrating that even this setting can significantly benefit from imagination-based

Table 2: The mean ± std and maximum success rate of DREMA trained on single-task demonstrations from different types of transformations over 5 test runs on 50 examples.

| | close jar | | sort shape | | slide block | |
|---|---|---|---|---|---|---|
| | mean ± std | max | mean ± std | max | mean ± std | max |
| Replay | 10.0 ± 3.10 | 16 | 1.2 ± 0.98 | 2 | 26.0 ± 0.00 | 26 |
| Object Rotation | 25.2 ± 0.98 | 26 | 10.4 ± 2.65 | **14** | 42.0 ± 0.00 | 42 |
| Roto-translation | 41.2 ± 2.40 | 44 | 10.0 ± 2.53 | **14** | 50.4 ± 1.50 | 52 |
| DREMA + Original *(All data)* | **51.2 ± 1.60** | **54** | **11.2 ± 1.60** | 12 | **62.0 ± 2.19** | **66** |

demonstrations. DREMA trained on *only imagination demonstrations* and evaluated in the original environment still performs significantly better than PerAct (+3.1 on single task +9.1 on multi-task).

## 5.2 ANALYSIS AND ABLATIONS

**Varying dataset size (Figure 4 & Figure C.6).** We analyze the effectiveness of DREMA's imagined demonstrations based on the number of original demonstrations provided. Specifically, we vary the dataset size $|\mathcal{D}| \in \{4, 10, 20\}$ for the *slide block* and and $|\mathcal{D}| \in \{5, 10, 20\}$ *sort shape* tasks (see App. L for training details). Figure 4 shows the average success rate of DREMA and PerAct as a function of the original dataset size. Increasing the training examples to 20 improves performance in both tasks. PerAct trained on 20 examples matches DREMA trained with imagined data from just 5 demonstrations (81 and 61 imagined demos for *sort shape* and *slide block*, respectively). However, when using the same 20 examples to generate imagined data (437 and 391 imagined demonstrations for *sort shape* and *slide block*, respectively), the model improves by 6.4% and 10.4%. This shows that generating novel configurations and actions remains beneficial, even with more demonstrations. Figure C.6 in the appendix shows how augmented data cover a wider portion of the robot's workspace, enabling the agent to handle configurations beyond those in the original demonstrations.

**Ablating transformations (Table 2).** We compare the equivariant transformations and include a baseline called *Replay*, which replays the original demonstrations. The *Object Rotation* and *Roto-translation* augmentations generate entirely imagined data. Both *Object Rotation* and *Roto-translation* increase the coverage of the state space and improve the generalization of the learned policy, with *Roto-translation* having the greatest impact. Combining these transformations results in even more general policies, yielding a 10.4% improvement in mean accuracy. This highlights the importance of learning a good world model and using it to explore and apply valid data augmentations.

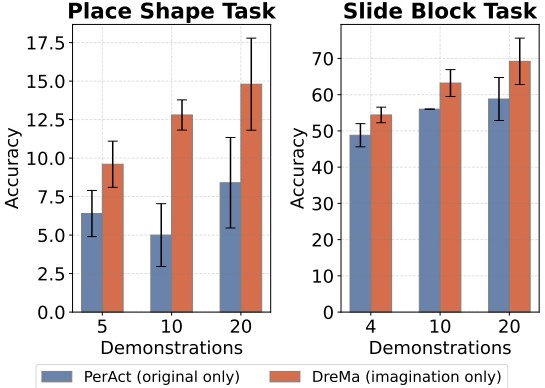

Figure 4: Imagined demonstrations keep improving imitation learning even with increasing number of original data.

## 5.3 EXPERIMENTS WITH A REAL ROBOT

**Real robot setup.** We verify the effectiveness of our approach in real-world environments. Our setup includes a Franka Emika Panda on a table, with a calibrated Kinect Azure mounted on the end-effector and a Kinect V2 positioned in front of the table, replicating the setup used in PerAct. We collect five tasks (number of demonstrations in Table 4). We train two models per task —one using only the original demonstrations and another combining original and imagined demonstrations— and do model selection as in PerAct (see App. L for details). The models are tested in 20 different environment configurations: 10 with object positions similar to the training set and 10 with positions outside the distribution (OOD), to assess generalization.

**Verification of the learned world model (Table 3).** To assess the accuracy of DREMA in modeling demonstrations, we compare the final object positions at the end of the demonstration

Table 3: Localization errors.

| Task | Error (m) |
|------|-----------|
| *pick block* | 0.010 |
| *pick shape* | 0.050 |
| *push* | 0.049 |
| Average | 0.038 |

Table 4: In- and out-of-distribution evaluation with real robots.

| | pick block 4 examples | | pick shape 5 examples | | push 3 examples | | place obj 4 examples | | erase 4 examples | | Average | |
|---|---|---|---|---|---|---|---|---|---|---|---|---|
| | In distr. | OOD | In distr. | OOD | In distr. | OOD | In distr. | OOD | In distr. | OOD | avg. | OOD |
| PerAct | 55 | 50 | 30 | 10 | 40 | 10 | 20 | 10 | 30 | 20 | 31.7 | 25.0 |
| DREMA (All) | **90** | **90** | **35** | **30** | **80** | **60** | **65** | **40** | **50** | **50** | **62.9** | **58.3** |

Figure 5: Original (top) and imagined demonstration (bottom) after a $90°$ rotation transformation.

with those predicted by the world model. Results show that the world model aligns well with the collected data for simpler demonstrations. For more complex examples, errors could increase. This discrepancy could be addressed using real demonstrations to adapt the world model's parameters.

**Real robot results (Table 4 & Table B.6).** The imagined demonstrations double the model's task performance (62.9% for DREMA vs 31.7% for PerAct on average). Moreover, PerAct, trained only on original demonstrations, struggled when the object distribution differed from the training data. In contrast, DREMA performs robustly in- and out-of-distribution environment configurations. The *pick shape* task had the lowest accuracy, likely due to the small size of the shapes and the model's difficulty in accurately recognizing them. Finally, in Table B.6, we also confirm that DREMA with *only imagined trajectories* can outperform original PerAct training ($+25\%$ on *pick block* task) when more examples are available, showing the usefulness of the imagined trajectories on their own.

## 6 CONCLUSIONS

We introduce a novel framework for constructing manipulation world models. By integrating real-time photorealism with Gaussian Splatting and physics simulators, DREMA enables robots to accurately predict the consequences of their actions and learn from minimal data. Our evaluations show significant improvements in policy learning, demonstrating the model's robustness and efficiency in real-world applications. We believe DREMA can influence other research areas, such as augmenting task variations with VLMs or enabling policy evaluation through imagination.

**Limitations.** Although DREMA demonstrates several strengths, it has limitations. First, it requires full observability of the environment to model physics and appearances. Second, it currently handles only simple objects, but recent advances in automatic URDF estimation could address this issue (see App. G). Third, we construct it with an open-loop approach without feedbacks; incorporating them into the world model construction could improve both prediction accuracy and policy learning. Finally, scaling the world model to large and diverse environments may increase simulation complexity. Overall, we believe that future research can address these issues, particularly the second and the third. Another interesting future direction is to study the effect of DREMA with more recent models (such as RVT2 (Goyal et al., 2024) or Diffusion Policy (Chi et al., 2023)) or approaches that directly exploit RGB images to show the benefit of Gaussian Splatting.

**Reproducibility statement.** We are committed to ensuring the reproducibility of our research findings. In our paper and in the appendix, we have provided comprehensive details about our methodologies, experimental setups, and hyperparameter configurations. In addition, details about inference and training runtime of both DREMA and corresponding agent trained with data generated by DREMA are presented in Appendix D. Furthermore, we will share the code and the synthetic datasets used for our simulation experiments. This will enable researchers to replicate our experiments and verify our results.

**Acknowledgments.** This work is supported by ERC Starting Grant EVA 950086. Andrii Zadaianchuk is funded by the European Union (ERC, EVA, 950086). Leonardo Barcellona was supported by a scholarship from Fondazione Ing. Aldo Gini (University of Padova) during part of the project.

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

# APPENDIX

## A    EXTENDED SINGLE TASK AND MULTI-TASK RESULTS

**Comparing with other augmentations.**   In the single-task setting, we compare DREMA to Per-Act trained with three invariant augmentation techniques: generating random patches on RGB-D images (Laskin et al., 2020), randomly altering the table color (Chen et al., 2023), and introducing random objects into the scene (Chen et al., 2023; Bharadhwaj et al., 2024). For each approach, we generate 80 augmented examples and combine them with the original data to train the policy.

To mask the table, we use ground-truth images provided by RLBench. Random distractors are introduced by embedding the point clouds of objects into the original re-projected point cloud and projecting them back onto the original views. The objects are randomly selected from 18 scans of the YCB dataset (Calli et al., 2015). We believe that combining invariant augmentations with equivariant augmentations has the potential to yield more robust and generalizable policies.

**Multi-task results.**   In multi-task settings, DREMA is tested on *close jar*, *slide block*, and *sort shape* simultaneously. The agent trained with both imagined and real data achieves the highest accuracy, except in the *sort shape* task, likely because shape matching is more refined and requires additional original training data.

Consistent with findings by Shridhar et al. (2023), we observe that while multi-task learning enables the agent to handle diverse tasks, it comes at the cost of lower accuracy for each individual task. The same three tasks achieved an average accuracy of 41.5% in the single-task setting.

Table A.5: Comparison of PerAct (Shridhar et al., 2023) trained on original demonstrations to DREMA trained on only imagination demonstrations and the combination of both in the multi-task setting. The table reports the mean ± std and maximum success rate over 5 test runs.

| | **Multi-task** | | | | | | | |
| | *close jar* | | *sort shape* | | *slide block* | | *avg multi-task* | |
| | mean ± std | max | mean ± std | max | mean ± std | max | mean | max |
|---|---|---|---|---|---|---|---|---|
| PerAct *(Original data)* | 26.0 ± 3.10 | 28 | 7.2 ± 1.60 | 10 | 34.0 ± 5.06 | 38 | 22.4 | 25.3 |
| DREMA *(PerAct with imagined data)* | 28.0 ± 3.35 | 32 | **18.0** ± 2.83 | **22** | 48.0 ± 1.79 | 50 | 31.3 | 34.7 |
| DREMA *(PerAct with all data)* | **46.0** ± 3.58 | **52** | 6.4 ± 3.20 | 12 | **54.0** ± 2.19 | **58** | **35.5** | **40.7** |

## B    IMAGINATION RESULTS IN THE REAL EXPERIMENT

In Table B.6, we present the complete results for the *pick block* task in the real-world experiment. This experiment aimed to verify whether the robot can learn purely through imagination, as demonstrated by the simulation results. We observe that with only four original examples, the learned policy performed worse than the initial policy. However, its performance improved significantly as more original examples were added.

Table B.6: Results of the *pick block* task with original and imagination demonstrations.

| | 4 examples | 8 examples |
|---|---|---|
| Real | 55 | 55 |
| Imagination | 50 | 80 |
| Imagination + Real | **90** | **85** |

## C    DATA GENERATION ANALYSIS

As mentioned in the work, a common problem of imitation learning, and particularly behavior cloning, is generalizing outside the given configuration of the environment. The ability of DREMA

to generate high-quality data by imagining new configurations of the environment allows us to increase the training distribution. Figure C.6 shows this effect by plotting the location of the keypoints (robot actions) in the original demonstration and in the generated imagination. This results in better and more robust policies. Gaussian Splatting guarantees that the domain gap is small while the simulator allows us to recreate a reasonable interaction between the robot and the objects.

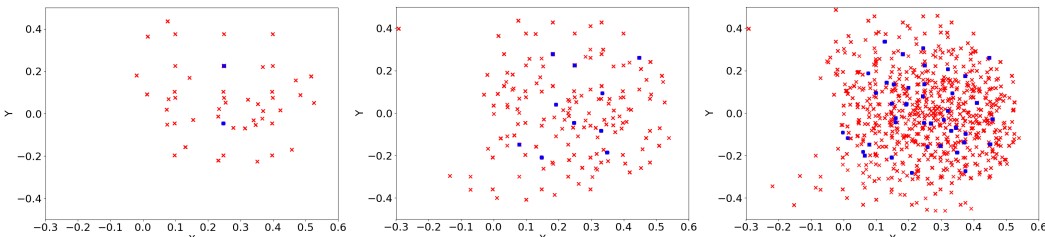

Figure C.6: Keypoints distribution of original (blue dots) and generated data (red cross) for the *sort shape* task. Data is generated from one examples (left), five examples (middle) and twenty examples (right).

## D  RUNTIME ANALYSIS

The data generation approach is performed offline, as its runtime scales linearly with the number of objects. Training PerAct in single-task configurations takes approximately two days on an Nvidia A40 for 100k iterations with batch size 4, so adopting an online approach would have minimal impact. Using an Nvidia RTX 4090, DEVA-tracking segments 5 images per second, while generating the Gaussian model and mesh for a single object takes about 4 minutes. Replaying trajectories is significantly faster, as it only involves applying roto-translations to the Gaussians. In our tests, the average time to reach a waypoint in tasks like *close jar*, *slide block*, and *sort shape* was 1.715 seconds without rendering and 1.883 seconds with rendering. Rendering required 0.168 seconds on average to update Gaussian positions, render five $128 \times 128$ RGB-D images, and filter them. The PerAct inference time is comparable to the one reported in the original paper (around 2 frames per second), as DREMA does not modify the model architecture but only generates training data.

## E  EXPLICIT WORLD MODELS AND LEARNABLE DIGITAL TWINS

Typically, an implicit world model (Ha & Schmidhuber, 2018; Hafner et al., 2019) would consist of the *encoder* that maps observations into a latent state, a *latent dynamics prediction* (including reward prediction) and a decoder to reconstruct original observations for a latent state. Using such a model after training, an agent can imagine the possible future trajectory and use it to learn a policy.

In our work, the encoder is a neural network masking the RGB inputs, mapping onto a Gaussian Splat Object Asset representation. Gaussian Splats can be seen as shallow (1-layer) Neural Field networks. The decoder is the Gaussian Splatting reconstruction, per object.

The representation function of the world relies on (i) the Gaussian Splat Object Assets, and (ii) the physics simulator to generate, aka "imagine", new RGB-D representations of novel future world states never seen before. We can use the imagined states to train our policies. This aligns with Ha and Schmidhuber's idea of an agent being trained entirely within "its own hallucinated dream" (Ha & Schmidhuber, 2018).

Similarly, in our experiments, the PerAct agent was trained using imagined sensory data (RGB and depth images rendered via Gaussian Splatting). The results in Table 1 demonstrate the agent's ability to use this generated data, while Table 4 highlights its effectiveness in a real-world setting. By integrating this generated data with the original data, we demonstrate improved performance, further validating the predictive power of the constructed world model.

In summary, our approach satisfies key components of an *explicit world model*:

- **Latent State**: Represented as implicit latent vectors in traditional models, while in our case, it corresponds to explicit compositional representations that consist of the position of Gaussian splats, meshes, and other environment parameters.

- **Observations Reconstructions**: Traditionally inferred by a neural network, here derived from Gaussian Splatting renderings.

- **Dynamics Prediction**: Traditionally learned by reconstruction of the observation, while in our work, constructed using explicit representation and physics engine.

**World models and learnable digital twins.** We further highlight the connection between world models and learnable digital twins, as in Abou-Chakra et al. (2024). When an agent learns a digital twin from observations and exploits it for decision-making or policy learning, the boundary between the two becomes blurred. Learnable digital twins can be seen as explicit models of the environment. While standard world models are typically implicit, both they and the more explicit digital twins serve the shared purpose of enabling agents to simulate and reason about their surroundings. This connection is further supported by Yang et al. (2023), which uses implicit and interactive world models as simulators.

In conclusion, while our work incorporates elements of real2sim and data augmentation, we believe it fundamentally adheres to and extends the principles of world models. The growing overlap between these areas underscores the convergence of these methodologies, and we believe our contribution aligns with this evolution.

## F  TASKS DESCRIPTIONS

We give more details about the imitation learning tasks described in the experiments. The **simulation tasks** are chosen from those defined in PerAct by Shridhar et al. (2023) without modifications (see Figure F.7 and Figure F.8). The **real tasks** have been designed to be similar to those used in the simulation (see Figure F.9).

### F.1  SIMULATION TASKS

**Close jar.** Place the lid on the jar with the specified color, sampled from a set of 20 color instances. In each scene, there is a target lid and a distractor lid. The task is considered correct when the robot releases the lid on the jar.

**Insert peg.** Insert a small square inside a colored peg. The size of the square has been slightly increased to improve reconstruction. The task is successful when the square is inside the correct peg.

**Lift.** Pick a colored cube identified by the prompt and move it to the red spot. Since the colored spot is not collidable in RLBench, it was manually set as non-collidable. The task is successful when the correct cube reaches the target.

**Pick cup.** Pick a target cup from two color variations and lift it. The task is successful when the correct cup is lifted.

**Sort shape.** Pick the prompted shape and place it inside the correct hole in the sorter box. In each scene, there are four distractor shapes and one correct shape to place. The task is considered correct when the correct shape is inside the box.

**Place wine.** Pick a bottle and place it on the rack. The task is successful when the bottle is released in the correct location on the rack.

**Put in cupboard.** Pick an object identified by the prompt and place it inside a cupboard. Since the cupboard is floating in RLBench, it was manually set as non-collidable.

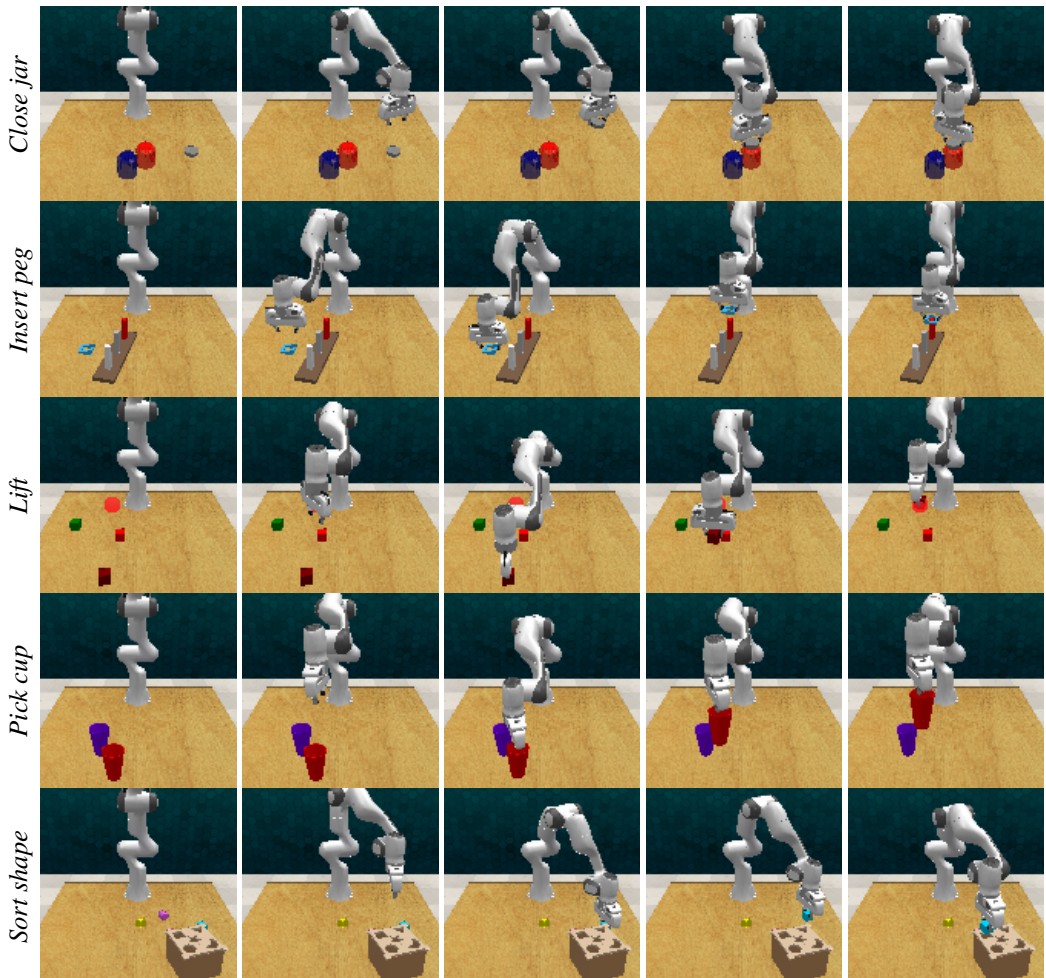

Figure F.7: Visualization of the simulation tasks. From top to bottom: *close jar*, *insert peg*, *lift*, *pick cup*, *sort shape*

**Slide block.** The task involves sliding a red block onto one colored spot. The target colors are limited to red, blue, pink, and yellow (four variations). The task is considered successful when part of the block is inside the specified colored area.

**Stack blocks.** Stack a given number of blocks with colors selected by the user. The task is successful when the desired number of blocks are correctly identified and stacked.

## F.2 REAL WORLD TASKS

**Pick Block.** The task consists of picking a green block and putting it on top of one bigger blue or red block. In each scene, both the possible targets are on the table. The task is considered correct if the robot releases the green cube on the right block and touches the correct place. We do not require the green block to stay on the block since we do not wish to show high accuracy in position, but the robot learned to distinguish the two correct positions. Training demonstrations: the cube is never moved from the position while the two target blocks are equally distant from it and symmetric to the $x$ axis of the robot base. We swapped the two blocks in each recorded demonstration. Test out of domain: the blocks are placed asymmetrically with respect to the $x$ axis, and the cube is placed in a random position.

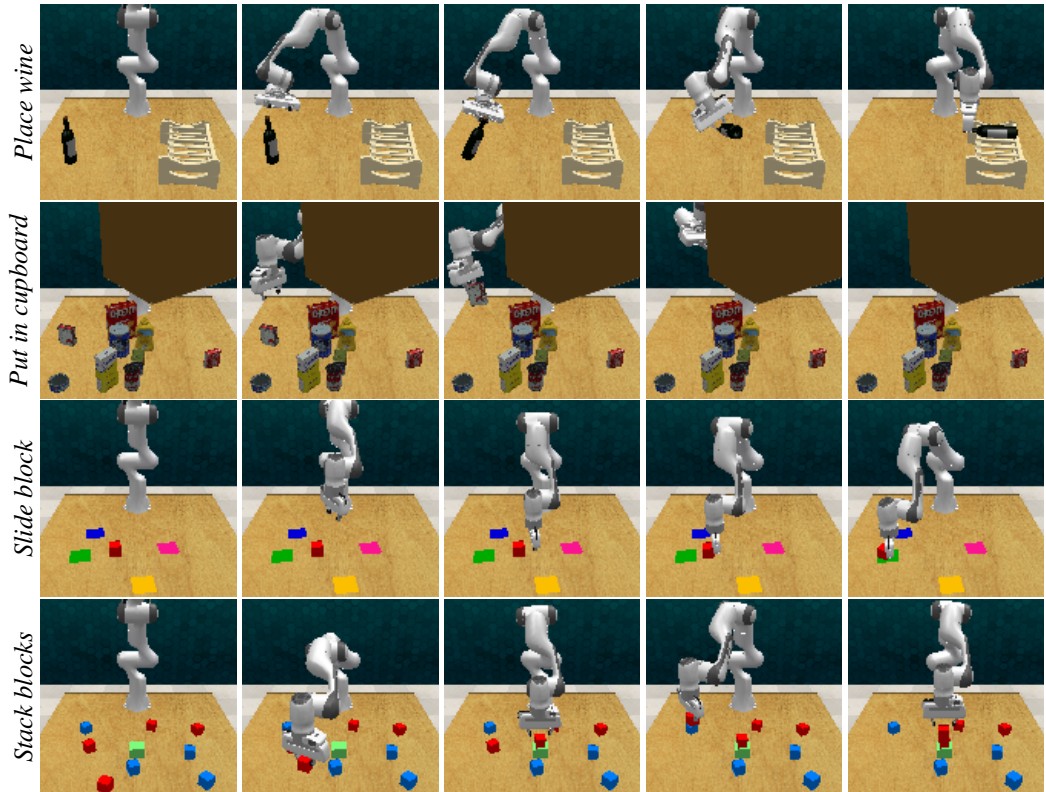

Figure F.8: Visualization of the simulation tasks. From top to bottom: *place wine*, *put in cupboard*, *slide block*, *stack blocks*

**Pick Shape.** The task consists of picking a colored shape and putting it inside a red box. In each scene, there is an orange star, a blue cube and a pink cross. All the shapes are visible in the scene. The task is considered correct if the robot releases the right shape inside the red box. Training demonstrations: The shapes are randomly placed in the scene, while the red box is always on the left side of the robot, close to the base. Test out of domain: the shapes are randomly placed in the scene, but the box is positioned on the other side or in a position far from the (fixed) training one.

**Push.** The agent is required to pick a screwdriver and use it to push a green cube. We considered the task correct if the robot touched the green cube with the screwdriver. Training demonstrations: the cube is never moved from the position while the screwdriver is placed on the left, on the right and in front of the cube. Test out of domain: the blocks and the screwdriver are randomly placed in the scene.

**Place Object.** The task consists of picking an object selected by the user and put it inside a box. In each scene, there is a tape, a stapler and a yellow duck (that is a distractor). All the shapes are visible in the scene. The task is considered correct if the robot releases the right object inside the brown box. Training demonstrations: The objects are randomly placed in the scene, while box is always on the left side of the robot. Test out of domain: the objects are randomly placed in the scene, but the box is positioned on the other side or in a position far from the training one.

**Erase.** The agent is required to pick an eraser and use it to erase a colored spot among two variations. We considered the task correct if the robot can erase slide the object on the right color. Training demonstrations: eraser is in the middle and the two colored spot in a fixed position. Test out of domain: the eraser is randomly placed.

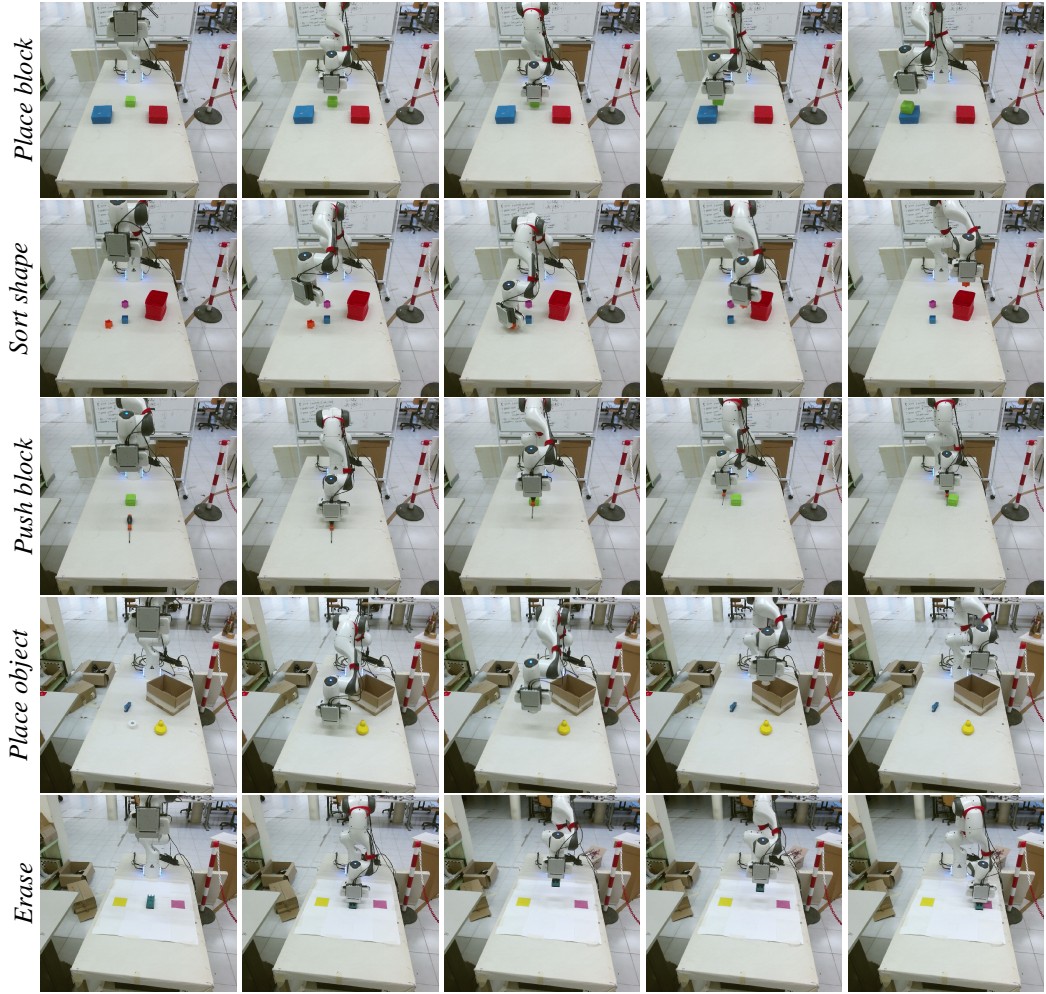

Figure F.9: Visualization of the real tasks. From top to bottom: *place block*, *sort shape*, *push block*, *place object*, *erase*.

## G   EXTENDING THE WORLD MODEL TO ARTICULATED OBJECTS

Automatic modeling of articulated objects remains an open challenge (Kerr et al., 2024; Liu et al., 2023; Weng et al., 2024). While DREMA does not currently identify articulated objects, it does support their simulation (for instance, the robotic arm itself is an articulated object). To model articulated objects, three key steps are required: (1) recognizing the object's components, (2) identifying joint types and parameters (e.g., rotational about a point or translational along an axis), and (3) determining the joint's location. Once these aspects are defined, a URDF can be generated for simulation (Chen et al., 2024b).

Several promising directions could help address these challenges. One possible approach would leverages a robot's or human motion to infer object movement and joint properties, as demonstrated by Kerr et al. (2024) that used foundation models such as SAM (Kirillov et al., 2023) to identify object parts, while an optimization-based approach improves joint movement. Extracting the meshes of the parts and encoding their movement within a URDF would enable the automatic creation of articulated objects for DreMa. Alternatively, the articulated structure could be inferred directly from semantic information, as demonstrated by Chen et al. (2024b), where a model could be trained to reconstruct articulated objects directly.

In the current version, robot links are segmented manually, while the model is assumed to be given. Automating this process could be achieved by segmenting links using a foundation model (Kirillov et al., 2023) or a trained segmentation model for this task.

## H DISCUSSION ON FAILURE SOURCES

The current implementation of DREMA is highly dependent on the segmentation mask. As shown in Figure H.10, if the segmentation mask for the objects is incorrect, the robot cannot interact with them. Similar issues may arise from incorrect mesh predictions, which PyBullet uses to detect collisions. Inaccurate meshes can lead to unexpected collisions or missed collisions.

The verification process prevents generating incomplete or incorrect demonstrations. Incomplete demonstrations can result from singularities in the trajectory or from trajectories that require unattainable robot configurations. Incorrect demonstrations often occur when object rotations move them outside the table, causing them to fall.

When PerAct is trained with such faulty demonstrations, its performance is significantly impacted. Even a small number of problematic demonstrations can make the model incapable of executing the task.

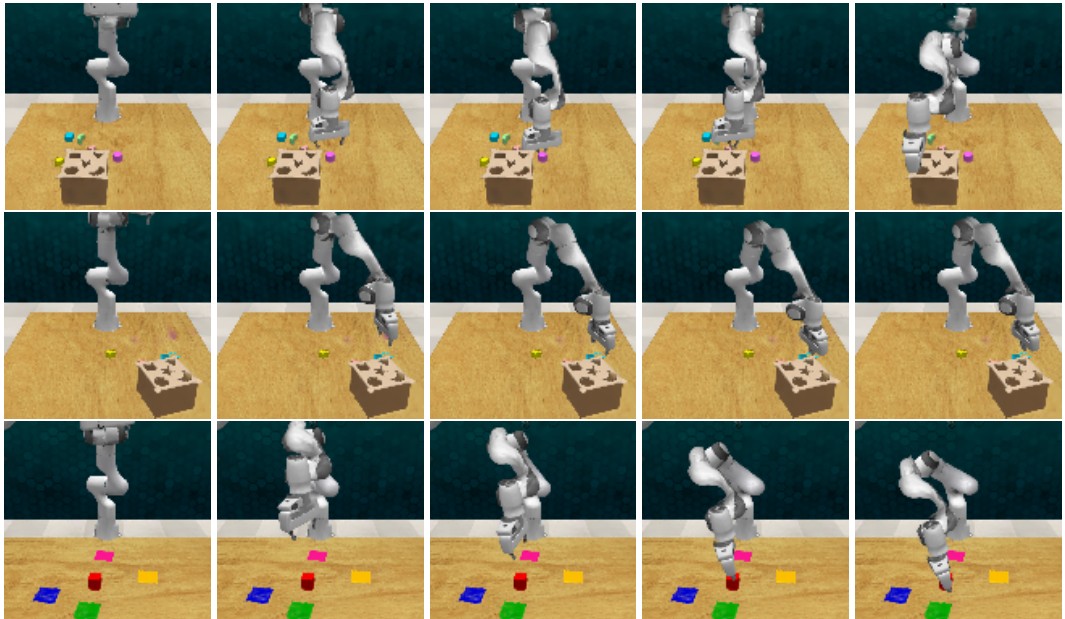

Figure H.10: From top to bottom consequences of wrong segmentation, consequences of wrong mesh prediction, consequence of no check of correct execution. In the first two rows the robot cannot pick the objects, while in the last row, it can't perform the actions they requires unreachable configurations.

## I EXTENDED RELATED WORKS FOR IMITATION LEARNING

### I.1 IMITATION LEARNING

Imitation learning involves collecting a dataset of demonstrations and learning a task from this data. Recent research has focused on vision-based imitation learning, where the policy is learned and executed directly from RGB or RGB-D images (Zeng et al., 2021; Shridhar et al., 2023; 2022; Goyal et al., 2024). A significant advancement in this area was Transporter (Zeng et al., 2021), which introduced a network for executing pick-conditioned placing. The authors analyze data augmentation and demonstrate its benefits for learning tasks, but argue that imagining various object configura-

tions is unrealistic. However, we show that generating new demonstrations is feasible, thanks to recent advances in foundation models, Gaussian splatting, and physics simulation.

CLIPort (Shridhar et al., 2022) adapts Transporter for multi-task object manipulation by incorporating CLIP features into the action computations. However, it is limited to simple pick-and-place tasks in 2D top-down settings. C2F-ARM (James et al., 2022) and PerAct (Shridhar et al., 2023) voxelized the environment and inferred actions directly in 3D. RVT (Goyal et al., 2023) builds on PerAct and uses the point cloud to render images from an optimal viewpoint to predict subsequent actions. All of these models require around one hundred demonstrations to learn a task. Recently, GNFactor (Ze et al., 2023) and RVT2 (Goyal et al., 2024) reduced the number of examples needed using NeRF and a coarse-to-fine inference, respectively. However, they still require many examples to perform tasks accurately. In this work, we demonstrate that by reconstructing a physics-powered simulation of the scene and imposing equivariant constraints on the data, we can generate new data and further reduce the number of demonstrations needed.

## I.2    DATA AUGMENTATION FOR IMITATION LEARNING

Data augmentation is a well-established practice in Computer Vision. Early milestone works (Krizhevsky et al., 2012; He et al., 2016) applied random flipping and cropping to improve model robustness. However, data augmentation in robotics presents unique challenges, as the robot's actions directly affect the environment. Thus, augmenting data is not always straightforward. Laskin et al. (2020) demonstrated that image augmentations, such as cropping or color jittering, can help agents learn better policies. However, their approach does not address augmenting robot actions. In contrast, our approach generates new trajectories.

In MoCoDa (Pitis et al., 2022), data generation is achieved by creating counterfactual transitions between states and actions. Recently, Corrado & Hanna (2024) proposed dynamic invariant augmentations, showing that expanding the state-action coverage of the training data is more effective than increasing transitions.

The recently introduced MimicGen (Mandlekar et al., 2023) demonstrated that imitation learning can benefit from data augmentation by composing different object-centric segments. The main limitation of these approaches is that they can only generate data by composing known states. In contrast, our approach generates new data by imagining novel equivariant configurations of objects and trajectories.

## J    ENGINEERING THE WORLD MODEL

The proposed compositional manipulation world model is a significant undertaking, requiring careful engineering to ensure its utility with real-world robots. However, the advantage of this approach is that, with the compositional manipulation world model we have developed, we can learn novel policies on a real robot with just a single example. For transparency and reproducibility, we provide a detailed description of the key engineering efforts involved.

**Low-resolution rendering.**    While Gaussian Splatting is designed for novel view synthesis, we observed unexpected behavior with low-resolution images, such as those in RLBench (James et al., 2020). Initially, we collected images at the original resolution of $1280 \times 720$ and rendered them at $128 \times 128$, the resolution required by PerAct. However, this change in resolution led to incorrect renderings of RGB images, and, more notably, depth images with inaccurate distance estimates from the cameras. As a result, the robot incorrectly predicted the consequences of its actions. To address this issue, we simply render the images at a higher resolution and then down-sample them to the desired resolution.

**Point cloud re-projection from 2D Gaussian Spatting.**    PerAct uses depth images to re-project the point cloud in 3D. Unfortunately, we noticed that the depths rendered from the Gaussians representation along the edges of the robot and the objects were not accurate enough. The consequence was to have noisy point clouds that affected the learning of the agent. We solve this problem by filtering out Gaussians near edges using the radius outlier removal from Open3D (Zhou et al., 2018).

**Mesh extraction and physics engines.** As motivated earlier, to model the object assets and their dynamics, physics engines require objects to be represented by mesh grids and not Gaussian blobs. We therefore adopt the 2DGS Gaussian Splatting (Huang et al., 2024) with TSDF to extract high-quality meshes, since the original Gaussian Splatting does not guarantee that the Gaussians align with the surfaces. Sugar (Guédon & Lepetit, 2024) proposes additional regularizations to improve mesh extraction. However, we found that their meshes were not accurate enough for reliable simulations. As a physics engine, we use PyBullet as a core component.

**Flat surfaces.** To avoid falling objects in the simulator, the table is extracted by filtering the depth by using the segmentation mask. RANSAC (Fischler & Bolles, 1981) is used to estimate a plane necessary to create a fixed convex hull that functions as a playground for the robot.

**Camera Calibration.** Unlike the graphics literature, which primarily focuses on the visual quality of reconstruction, and physics simulators designed for general or manually specified scenes, our goal is to integrate automated scene reconstruction with physics simulators. This requires aligning the geometry of the scene reconstruction with the geometry of the physics simulator. We achieve this by leveraging the calibrated camera mounted on top of the robot (Allegro et al., 2024). Additionally, since the robot agent is aware of its own dynamics, we can use its kinematics to estimate the extrinsic camera parameters as the robot moves. For more details on the real experiment, see Sec. 5.3, and for further information on camera calibration, refer to App. M.

**End-effector manipulations.** To move the robot inside the simulator, we specify a target pose that uniquely identifies the position and orientation of the end-effector. The robot is then controlled in position until the target is reached. Using the robot's kinematic model, expressed in the URDF, we can locate the different links and update the Gaussian accordingly.

**Background.** Once we have the Gaussian representation of the objects, we then extract the representation of the table surface and the surrounding environment. This is done by removing the object Gaussians from the rest of the scene to avoid duplicates.

**Physical parameters.** For the physics simulator, the geometry of the objects is not enough; various physical parameters are also required. Some physical parameters can be assumed to be constant, such as the gravitational constant or air resistance, while others are not, including the masses of objects, object elasticity, or the friction coefficient. In general, estimating physical parameters is a whole exciting field on its own (Mehra, 1974; Chebotar et al., 2019; Gao et al., 2022; Huang et al., 2023) and some of these methods, such as ASID (Memmel et al., 2024) or AdaptSim (Ren et al., 2023), can be applied to identify them. In this paper, we focus on learning a photo-realistic representation of the world powered by a physics simulator, and its usefulness for compositional manipulation world models and imitation learning. Keeping in mind the overall complexity, we consider physical parameter identification to be out of scope, yet an important direction for future work.

## K OPEN-VOCABULARY SEGMENTATION IN SIMULATION AND IN REAL WORLD

Segmenting simulation images with Grounded-SAM (Ren et al., 2024) proved challenging due to the domain gap. While the typical gap occurs from simulation to the real world, in our case, we encountered the opposite issue. The open-vocabulary segmenter, which was primarily trained on real data, struggled to accurately detect objects, the table, and the robot in the simulation. We observed that while SAM could identify objects, grounded-DINO misclassified instances, assigning incorrect labels. Figure K.11 shows examples of predicted masks in the simple environment with a cube in the center. In contrast, using only simple prompts such as "table" and "object", and filtering out masks not on the table, the segmenter could correctly identify objects in the real data. Figure K.12 displays examples from the real setting. One possible solution to improve recognition in the simulation is to fine-tune the segmenter to align the two domains. However, since our goal is to develop a world model with real-world applicability, we chose to focus on enhancing the zero-shot capabilities for the real world.

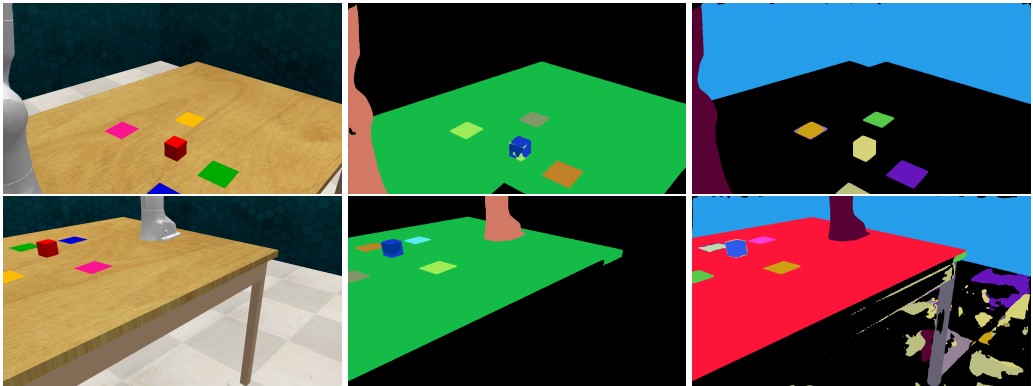

Figure K.11: Segmentation masks in simulation. The middle column was generated with the prompt "object", whereas the right column was generated with the prompt "green cube".

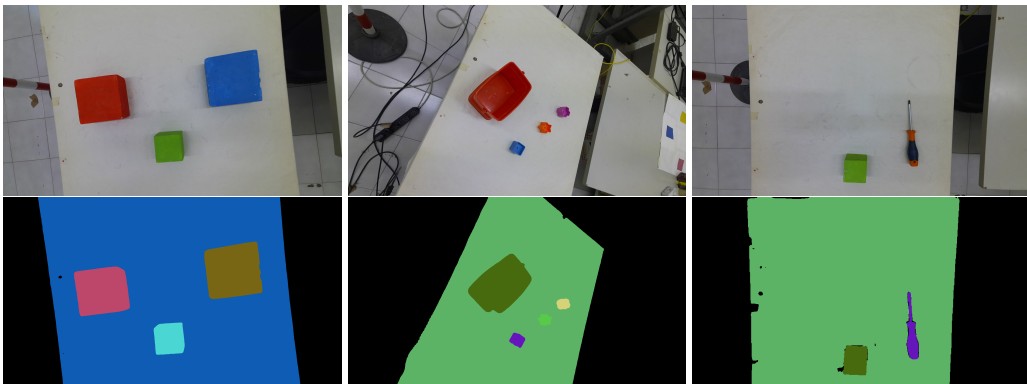

Figure K.12: Examples of segmentation masks from the real images. These were generated by prompting "object" and "table".

## L   WORLD MODEL LEARNING DETAILS

To avoid falling objects in the simulator, the table is extracted by filtering the depth using the segmentation mask. RANSAC (Fischler & Bolles, 1981) is used to estimate the plane necessary to create a fixed convex hull. Finally, the entire scene is reconstructed, and the Gaussians of the objects are removed from the reconstruction. The procedure produced a set of Gaussians paired with the mesh for each object and the Gaussians of the scene without the extracted objects.

**World Model Learning.**   The set of images $\mathbf{X}$ is collected using a similar approach of Ze et al. (2023) collecting 200 images from a rotating camera. We set the resolution of the camera to $1280 \times 720$, mimicking a real one. To avoid falling objects in the simulator, the table is extracted by re-projecting the depth filtered by the segmentation mask of this particular object. RANSAC (Fischler & Bolles, 1981) is used to estimate a plane, necessary to create a fixed convex hull used as a workspace. The Gaussians of each object are trained for 7000 iterations. The meshes are extracted starting from the Gaussians using TSDF and the hull. The table is used to remove the parts of the mesh penetrating the table. Without this filtering, PyBullet may detect collisions creating problems in the simulation. A similar operation should be also done in case of touching objects. The entire scene is reconstructed. The Gaussians of each object are matched with their counterpart in the entire scene using KNN and then removed to avoid duplications. The procedure produced a set of Gaussians paired with the mesh for each object, and the Gaussians of the scene without the extracted objects.

In imitation learning, the agent learns from a dataset $\mathcal{Z} = \{\zeta_1, \ldots, \zeta_M\}$ consisting of $M$ demonstrations, each paired with a corresponding task encoding $\mathcal{T} = \{t_1, \ldots, t_M\}$. When tasks are described

by language-based goals, the task $t_m$ is represented by a language embedding (e.g., pre-trained CLIP embeddings) corresponding to the phase that describes the task. Each demonstration $\zeta_m$ consists of a sequence of continuous actions $\mathbf{A} = (a_1, \ldots, a_t)$, which are encoded as end-effector poses and gripper states. These actions are paired with observations $\mathbf{Q} = (q_1, \ldots, q_t)$, which in our case can be the object assets from the compositional manipulation world model.

**Data Generation.** Following PerAct (Shridhar et al., 2023), we use a dataset $\mathcal{Z} = \zeta_1, \zeta_2, \ldots, \zeta_n$ of $n$ demonstrations, each paired with a corresponding task encoding $\mathcal{T} = t_1, t_2, \ldots, t_n$. For language-based goals, the task $t_i$ is the language embedding (e.g., a pre-trained CLIP embedding) of the phrase describing the task. Each demonstration $\zeta$ is a sequence of continuous actions $\mathbf{A} = a_1, a_2, \ldots, a_t$ paired with observations $\mathbf{Q} = (q_1, \ldots, q_t)$. An action $a$ consists of the 6-DoF pose, gripper open state, and whether collision avoidance was used by the motion planner to reach an intermediate pose: $a = a_{\text{pose}}, a_{\text{open}}, a_{\text{collide}}$. An observation $\tilde{q}$ consists of RGB-D images from any number of cameras. We use three cameras for simulated experiments, $\tilde{q}_{\text{sim}} = q_{\text{front}}, q_{\text{left}}, q_{\text{right}}$, and a single camera for real-world experiments, $\tilde{q}_{\text{real}} = q_{\text{front}}$. Each image in $\mathcal{Q}$ has dimensions $128 \times 128$.

To generate the data, the robot executes $\mathbf{A}$ in PyBullet and renders a new set of observations using simulated cameras located at the same positions as $\tilde{q}_{\text{sim}}$. When the robot successfully completes $a_i$, the corresponding simulated observation is rendered, and the robot proceeds to execute $a_{i+1}$. It is important to note that the set of cameras $q_{\text{front}}, q_{\text{left}}, q_{\text{right}}$ is not used for optimizing the Gaussian representations. Instead, the novel-view synthesis capability of Gaussian Splatting is utilized. At the end of the execution, the objects' locations are saved. This data is then used to verify if the actions in the new environmental configurations can successfully complete the task.

The new configuration of the environment is generated by translating and rotating objects and the trajectory, while keeping the robot in the same position. We translate the environment along the $x$ and $y$ axes using combinations of $[0.0, 0.15, -0.15]$. The $z$-axis is not augmented, as the test environments do not involve different table heights. However, such augmentation could be beneficial if the test environment includes varying heights. The rotation of the table and objects is performed around the position $[0.30, 0.0, 0.0]$, with a complete rotation around the $z$-axis in steps of $30°$. For the trajectory rotation, the step size is reduced to $20°$. For object picking, no specific operations are applied since the demonstration provides the correct picking locations. However, reconstruction errors could impact the simulation and prevent correct object picking.

In single-task settings, PerAct is evaluated on nine tasks. For these tasks, DREMA generates $n$ valid examples, where $n$ is as follows: 112 for *close jar*, 136 for *insert peg*, 142 for *lift*, 142 for *pick cup*, 81 for *sort shape*, 36 for *place wine*, 58 for *put in cupboard*, 61 for *slide block*, and 63 for *stack blocks*. These demonstrations correspond to cases where the goal position was correctly reached in the augmented environment, as verified in Sec. 4. Each original demonstration allows for a maximum of 37 generated trajectories, with 37 original demonstrations used across the nine tasks. This results in a theoretical maximum of 1369 demonstrations, but filtering reduces this to 831, retaining approximately 60% of the generated data.

We noticed that the jars and shape sorter are fixed in the simulator, so we also fixed them in the experiments. Similarly, the cupboard, which is floating in the simulator, was adjusted accordingly in the reconstruction. Our goal is to create a representation that the robot can manipulate and use to learn tasks that visually resemble the real-world scene.

**Training.** In the single-task setting, PerAct is trained using the same parameters as the original implementation, except for a reduced batch size of 4 and a decreased number of iterations to 100k. We also trained PerAct on *slide block*, *close jar*, and *sort shape* tasks with the original batch size and found minimal differences in accuracy compared to the reduced batch size (*close jar*: +1.5, *slide block*: +1.2, *sort shape* : -2.8). For the multi-task setting, we follow the single task setting while increasing the number of iterations to 600k. All training and testing experiments with PerAct are conducted on an Nvidia A40 GPU.

**Validating and Testing.** Model selection is crucial to avoid deploying suboptimal policies (Shridhar et al., 2023). Compared to the original study, the number of validation examples was increased from 25 to 40 to improve the selection of models. Similarly, the size of the test set was increased from 25 to 50 to accommodate a wider range of evaluation scenarios. In addition, the test was re-

peated five times to incorporate the variability introduced by motion planning and environmental interactions.

**Rendering problems.** Gaussian Splatting is designed for novel view synthesis. However, we noticed strange renderings when it is trained with high-resolution images and deployed to generate low-resolution ones, that are needed for the experiments in RLBench (James et al., 2020). Figure L.13 shows how these problems affected the rendering of RGB and depth images. In particular, depth images must be carefully estimated since PerAct uses them to create a voxel representation of the environment. The rendering problem caused the robot to predict the actions incorrectly. To overcome this problem, we rendered a higher-resolution image and down-sampled it to the desired resolution.

**Point Cloud re-projection from 2D Gaussian Spatting.** PerAct uses depth images to re-project the point clouds in 3D. We noticed that the depth images rendered by the Gaussians are inaccurate along the edges of the robot and the objects. This created noisy point clouds that affected the agent. Figure L.14 shows a point cloud re-projected from the generated depth. We solve it by filtering using the radius outlier removal from Open3D (Zhou et al., 2018). This was particularly effective with far cameras but less effective with the wrist camera. Therefore, we used three cameras instead of four for the experiments.

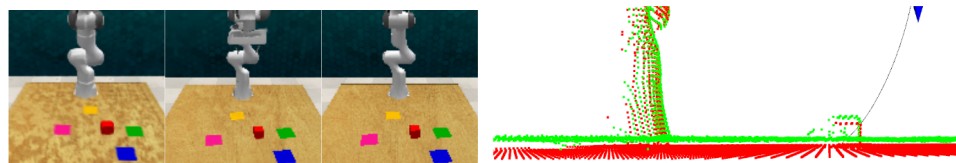

Figure L.13: Three left images show the rendered RGB, the same RGB rendered with a higher resolution, and the original image. On the right, the original point cloud (red) does not align with the rendered one (green) when the image is directly rendered in low resolution.

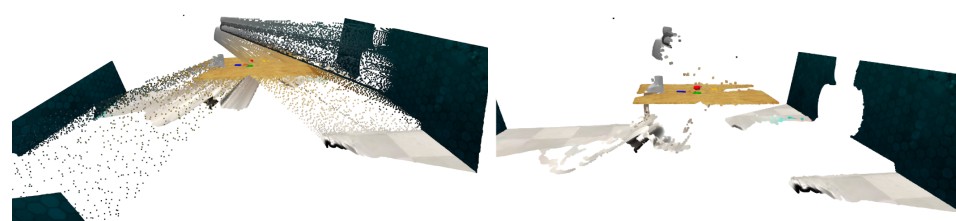

Figure L.14: Re-projected point clouds before (left) and after (right) the filter.

## M    MORE INFORMATION ON THE REAL EXPERIMENT

In this section, we provide a more detailed description of how we learned the world model and deployed imitation learning in a real robot setup. Moreover, we add some brief discussions on problems we encountered.

**Collecting demonstration in the real world.** One of the most essential parts of imitation learning is data collection. At the beginning and the end of the demonstration, we recorded the images used to reconstruct the world model and checked that with the physical parameters used, the robot could replay the same trajectory. The demonstrations are collected only from the external Kinect V2 camera, positioned to resemble the *front* camera of RLBench as in the real experiment of PerAct (Shridhar et al., 2023). The keypoints to execute were manually saved through the RVIZ interface in ROS1 (Quigley et al., 2009). When the keypoints were established, we used sampling-based motion planning to generate the execution trajectory. During the execution, we recorded a Ros bag

later processed to create data in RLBench format[1]. In collecting the data, we noticed that quality is essential for correctly training the model. Sometimes formatted data contained more keypoints in similar positions, creating loops in the agent that could not exit. We encountered problems when the same task in a similar position had a different number of keypoints. We believe that these problems are reduced by increasing the batch size, but due to time constraints, we could not verify this assumption. To avoid these issues, we collected demonstrations with a constant number of keypoints, and manually cleaned the data when we noticed unwanted keypoints.

**World model creation and data generation.** The images needed to construct the world model were collected with the camera on the robot. The camera was calibrated, and the inverse kinematics was used to retrieve the camera pose. However, we noticed that Gaussian Splatting requires accurate poses. Sometimes, the movements of the robot or the interactions with the objects made the camera move. Consequently, we had to calibrate the camera again. The alignment between the world model and the real world played another important role. We observe that this should be accurately estimated. Otherwise, the replayed trajectory would miss the objects.

**Training and testing PerAct.** We trained PerAct with the original parameters but with a batch size of 4 for 100k iterations. This was constrained by the training time required since we wanted to test policies with different training data. The original implementation saves a model every 10k iterations. We reduced this to 5k to keep more models close to the final part of the training. Model selection in the real world was not straightforward. We noticed that choosing the wrong model highly impacts the accuracy. While collecting a validation set and checking the loss is possible, this is not the best solution because the only way to understand the best model is by making the agent interact with the environment. Consequently, we tested the last five models with a minimal number of configurations. We think that model selection in the real world is a crucial problem that opens the way to possible future works for our world model.

---

[1]https://github.com/baldeeb/rosbag_data_utils/tree/main

# N EXAMPLES OF GENERATED DATA

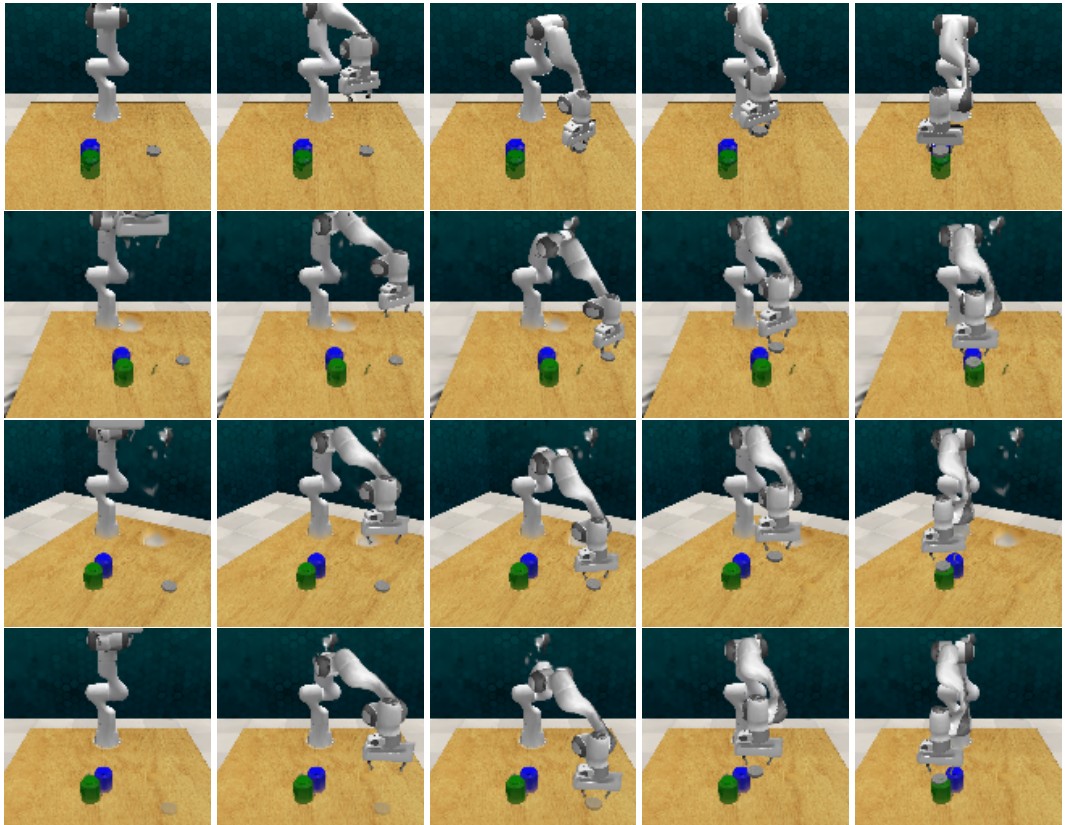

Figure N.15: Visualization of the *close jar* task

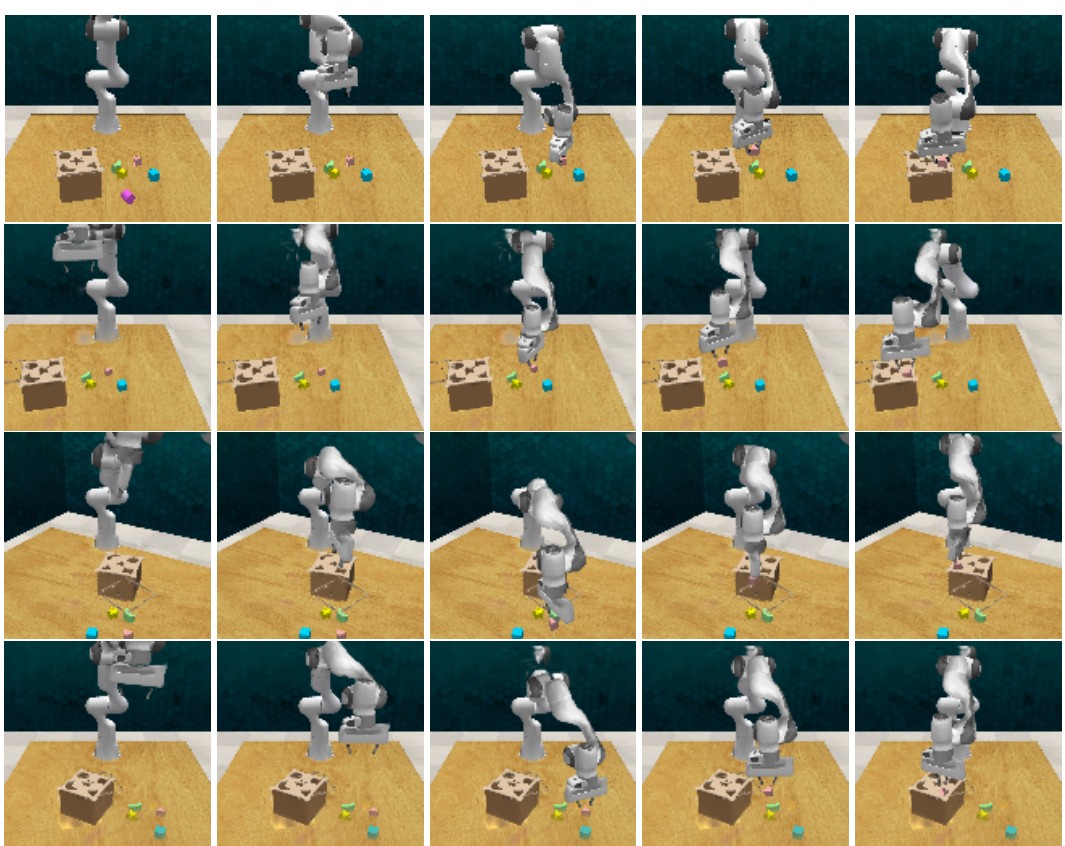

Figure N.16: Visualization of the *sort shape* task

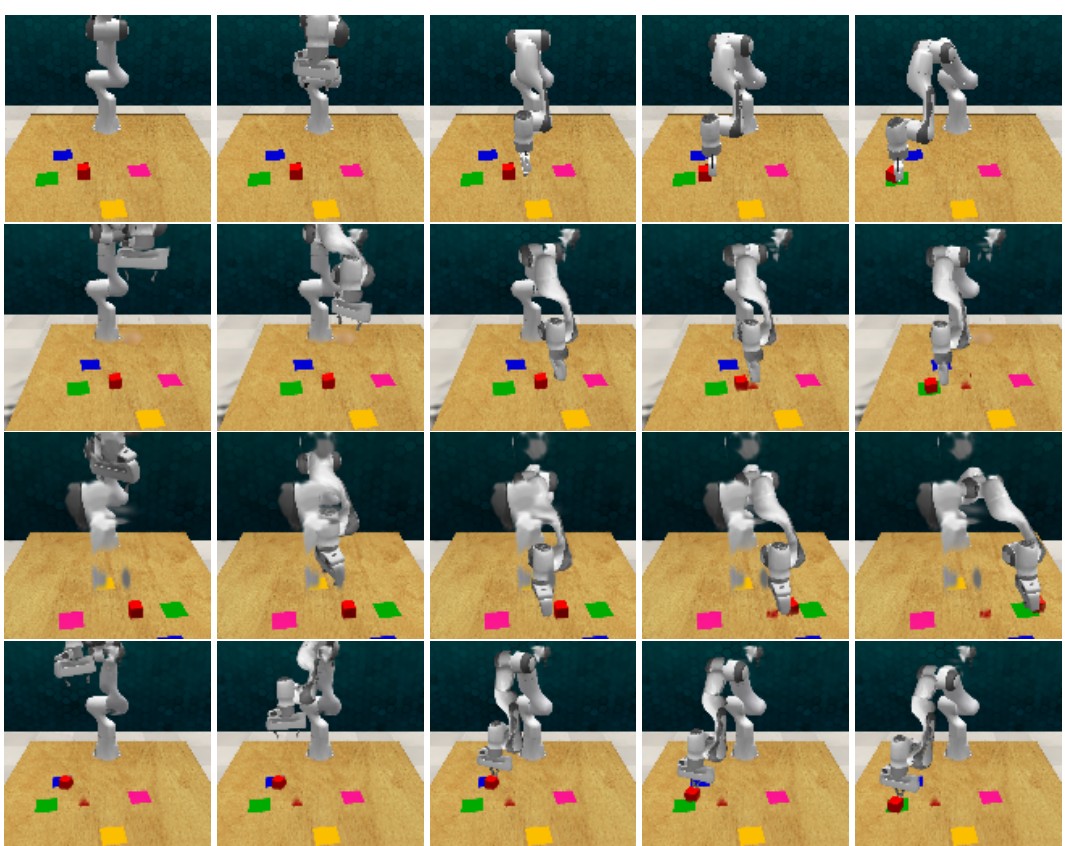

Figure N.17: Visualization of the *slide block* task

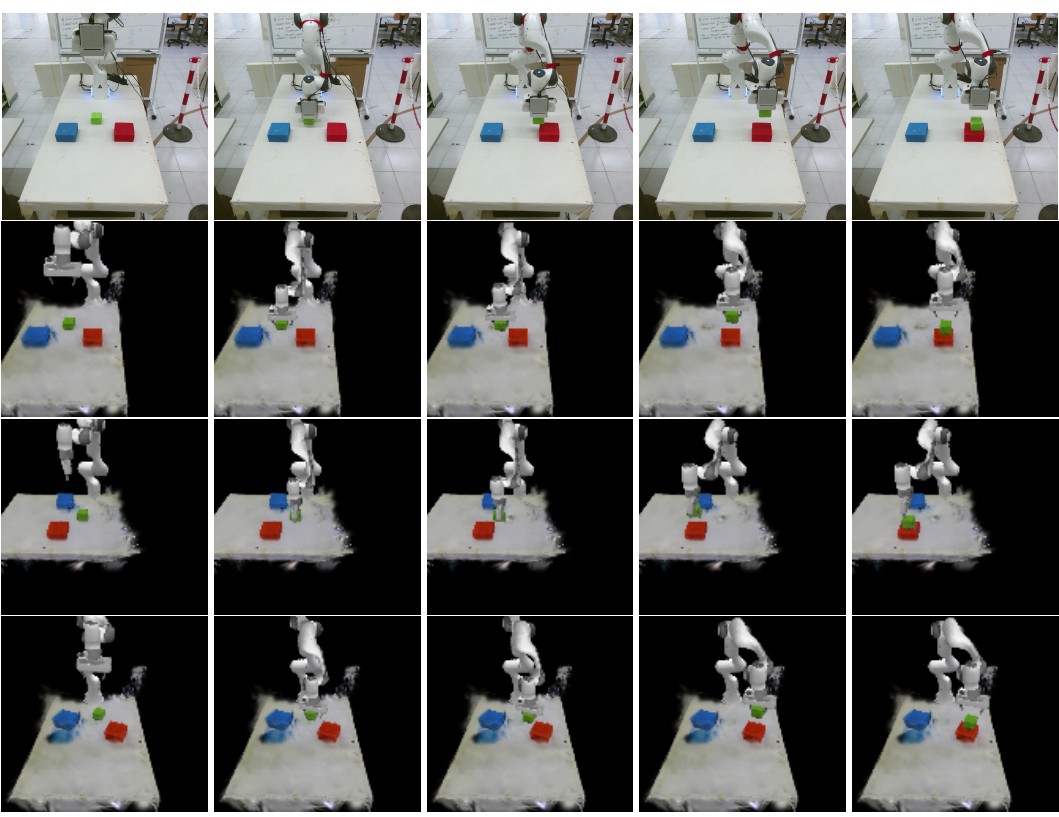

Figure N.18: Visualization of the *pick block* task

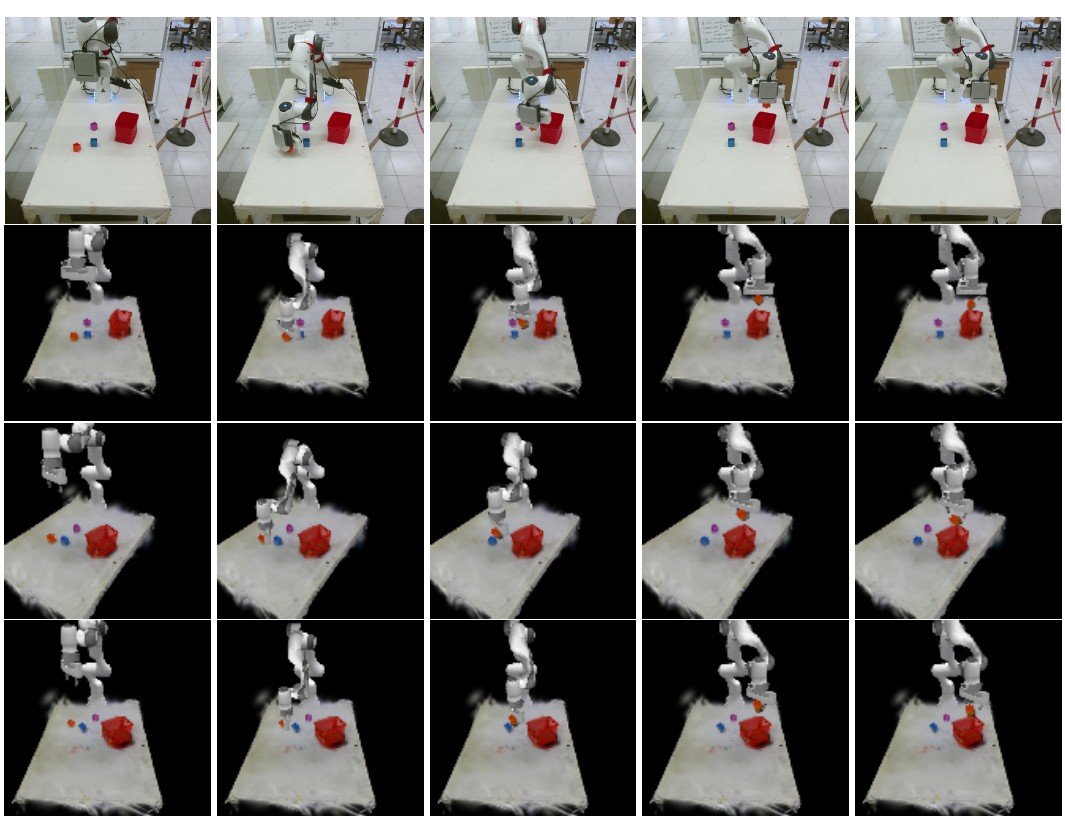

Figure N.19: Visualization of the *pick shape* task

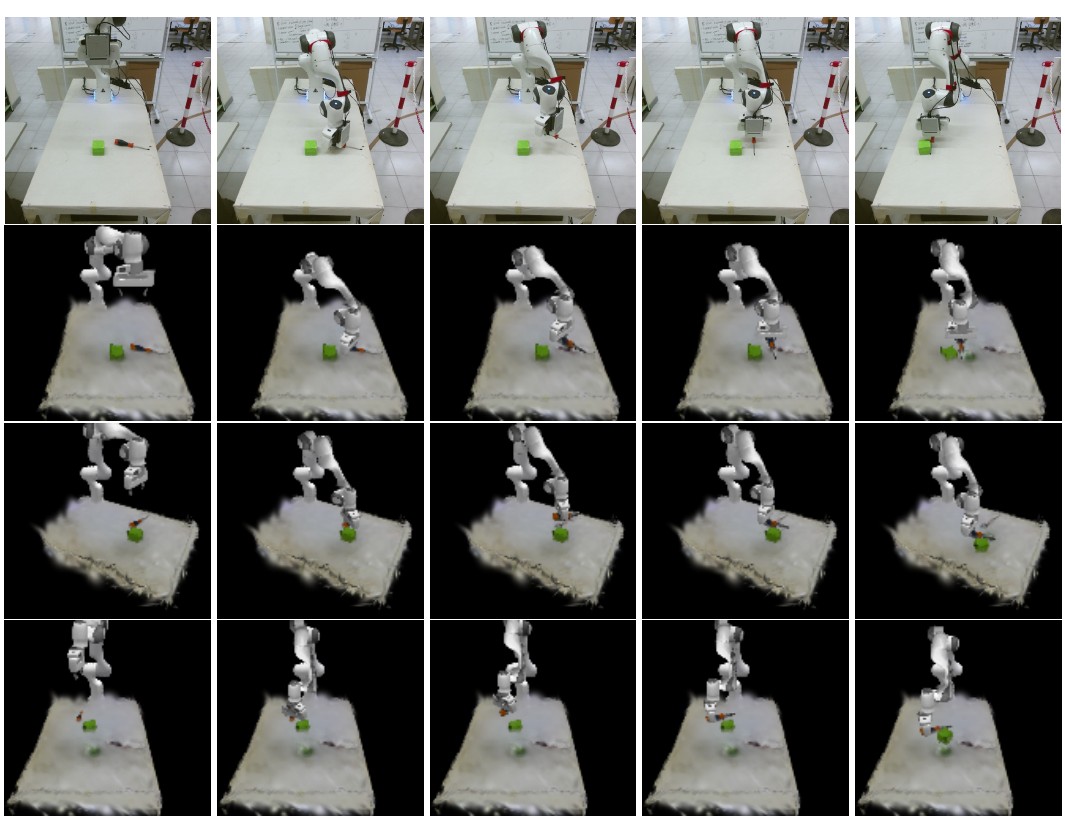

Figure N.20: Visualization of the *push* task

