# OpenReview forum: "Dream to Manipulate: Compositional World Models Empowering Robot Imitation Learning with Imagination"
_ICLR.cc/2025/Conference — ICLR 2025 Poster_

### Official Review · Reviewer_Mne8 · 2024-10-20

**Soundness:** 3
**Presentation:** 2
**Contribution:** 3
**Rating:** 8
**Confidence:** 4

**Summary:**

The paper presents a novel paradigm for constructing world models that serve as explicit representations of real-world environments and their dynamics. By integrating advances in real-time photorealism, such as Gaussian Splatting, with physics simulators, the authors propose a system capable of generating new data for imitation learning. Additionally, the paper demonstrates the application of this model in real-world scenarios, showing how data collected from the world model can be used to train robots via imitation learning, with promising results when transferring learned behaviors to real-world tasks.

**Strengths:**

1. The paper introduces an innovative approach by leveraging world models to generate robotic data for imitation learning, which is a contribution to the field.
2. The experiments are detailed, covering both simulation environments and real-world robot demonstrations, providing a robust evaluation of the approach.
3. A creative method for augmenting data used in imitation learning is introduced, which could lead to improved learning efficiency.

**Weaknesses:**

1. The absence of publicly available source code limits the reproducibility of the results. It is suggested to release the code during the rebuttal stage.
2. Some figures in the paper need improvement, as the text in several instances is too small to read clearly.
3. The predictions demonstrated in the paper are limited to simple tasks and physics environments, and future work should focus on extending these predictions to more challenging tasks and complex physical simulations.

In conclusion, the paper presents a compelling framework that blends world modeling with imitation learning, but there are areas for improvement, particularly in terms of figure clarity, task complexity, and providing source code for reproducibility.

**Strengths:**

1. The paper introduces an innovative approach by leveraging world models to generate robotic data for imitation learning, which is a contribution to the field.
2. The experiments are detailed, covering both simulation environments and real-world robot demonstrations, providing a robust evaluation of the approach.
3. A creative method for augmenting data used in imitation learning is introduced, which could lead to improved learning efficiency.

**Weaknesses:**

1. The absence of publicly available source code limits the reproducibility of the results. It is suggested to release the code during the rebuttal stage.
2. Some figures in the paper need improvement, as the text in several instances is too small to read clearly.
3. The predictions demonstrated in the paper are limited to simple tasks and physics environments, and future work should focus on extending these predictions to more challenging tasks and complex physical simulations.

**Questions:**

Could you please show some performance results in more complex physical environments and challenging tasks? Even if they were unsuccessful, it would be helpful to see such results, even though they are not included in the paper.

---

> ### Author Response · Authors · 2024-11-22
>
> Dear reviewer,
>
> Thank you for your kind review and your valuable suggestions.
>
> > Some figures in the paper need improvement, as the text in several instances is too small to read clearly.
>
> We increased the font size in Figures 1, 2, and 3 to ensure the text is easy to read. Additionally, we corrected the typo in Figure 2 and merged “1. Observation” and “2. Open-Vocabulary Tracking” into a single step to emphasize that segmentation is performed on the sequence of images to produce consistent object masks needed for masking the same object used in Gaussian Splatting.
>
> > The absence of publicly available source code limits the reproducibility of the results. It is suggested to release the code during the rebuttal stage.
>
> Thank you for the suggestion, we will definitely upload the code as soon as possible. At the moment, we cannot upload the code yet due to legal constraints by university regulations until publication. After acceptance, we will certainly clean up the code and share it all, together with models and data, in GitHub. We also highlight that we tried to insert all the necessary information to reproduce the experiments. (Appendix F, J, L, and M).
>
> > The predictions demonstrated in the paper are limited to simple tasks and physics environments, and future work should focus on extending these predictions to more challenging tasks and complex physical simulations.
>
> > Could you please show some performance results in more complex physical environments and challenging tasks? Even if they were unsuccessful, it would be helpful to see such results, even though they are not included in the paper.
>
> Considering the magnitude of the work required and the fact that we also wanted to implement the algorithm with real robots, we did not have the capacity at the submission time to add more tasks.
> Upon the reviewer’s request, however, we expanded our evaluation to 9 RLBench tasks and 2 new real-robot tasks, while using the same set of transformations. DreMa significantly improves over baselines, showing its general applicability in generating useful training data for diverse manipulation tasks. Detailed results are in Table 1 and Table 4 of the updated manuscript.
> A good segmentation is extremely important to obtain reliable data to train the agent. When the segmentation is wrong, as discussed in Appendix H, DreMa is not capable of generating useful data, degrading the policy learned by PerAct.
> In addition, we aim to extend this work targeting articulated objects to consider more complex environments in the future. In appendix G  we explain how the methods could be extended to more complex scenarios.
> One example of a less successful and challenging task was the place cups task of RLBench. There the number of generated examples of DreMa was highly reduced by the model verification, producing less than 10 examples. This, added to the difficulty of the model in learning the task resulted in the same accuracy of the original model, making the augmentation ineffective in this case.

---

> > ### Comment · Reviewer_Mne8 · 2024-11-23
> >
> > My concerns have been addressed, but I'm sorry I can't improve my score further. Congratulations, I think your work is valuable.

---

> > > ### Author Response · Authors · 2024-11-23
> > >
> > > We sincerely thank you for your positive feedback. Your recognition of our work is greatly appreciated, and your insightful comments have helped us further strengthen our submission.

---

### Official Review · Reviewer_kqVG · 2024-11-04

**Soundness:** 3
**Presentation:** 4
**Contribution:** 3
**Rating:** 8
**Confidence:** 4

**Summary:**

The paper augments a small set of demonstrations with imagination to improve few-shot imitation learning in both RL bench and real-world robotic tasks. The imagination comes from learning compositional objects models through Gaussian Spatting and replaying demonstrations with varied objects poses in a simulator.

**Strengths:**

The paper presents a novel strategy of data augmentation by first acquiring object models and then leveraging simulations to ensure correct dynamics of imagined demonstrations instead of learning worlds models of both objects and dynamics simultaneously. The strategy is shown to have meaningful improvement on few-shot imitation performance in sufficient sim and real tasks. The approach is also thoroughly invested with ablations showing the significance of the imaging with roto-translation of original demos. The paper is well written with clear motivations and goals and sufficient results to support the claim.

**Weaknesses:**

1. The paper should compare to other data augmentation approaches such as MimicGen or Digital Twin, which is significant extensive line of work worthy of more elaboration and discussion. Similar approach such as [1] should be discussed or cited.
2. The method relies on open-vocabulary tracking models to segment objects, which limit the approach to non-articulated objects. It is unclear how such segmentation models can capture individual robot link or object parts connected with articulated joints accurately. Also, it is unclear how to extend the simulator to incorporate articulated objects interaction after the parts have been learned.
3. Existing approach of verify imagined demonstration is rudimentary.

[1] MIRA: Mental Imagery for Robotic Affordances

**Questions:**

1. What’s the relationship with digital twin line of work?
2. How to learn 3d models for parts of articulated objects and how to imagine demonstrations that manipulate articulated objects?
3. Can you replay imagined trajectories in simulator to verify correctness? Will that help improve imitation success?
4. How does error build up in the pipeline of segmenting object masks -> learning objects models through Gaussian Spatting -> imagine demonstrations -> learning policy? Perhaps some ablations or quantitative metrics to measure error will be useful!
5. Can you add one more baseline method of data augmentation to compare to?

---

> ### Author Response · Authors · 2024-11-22
> **Part 1**
>
> Dear reviewer,
>
> Thank you for your valuable suggestions and for highlighting both the strengths and identify some concerns.
>
> > The paper should compare to other data augmentation approaches such as MimicGen or Digital Twin, which is significant extensive line of work worthy of more elaboration and discussion. Similar approach such as [1]( [1] MIRA: Mental Imagery for Robotic Affordances) should be discussed or cited.
>
> Thank you for your suggestion regarding additional related work. We added the comparison to MIRA in the uploaded version L148  and other related works connected to Digital Twin literature, such as [1] in L128 and [2] in L102. Mira exploits novel view synthesis from NeRF to improve the decision of the agent, while we use Gaussian Splatting to generate novel configurations of the environment. While MimiGen is indeed a data augmentation method, it requires a division of tasks into subtasks and the application of a pose estimator to track objects during execution (Mandlekar et al. 2023 Section 3).
> [1] Li, Xuanlin, et al. "Evaluating Real-World Robot Manipulation Policies in Simulation." arXiv preprint arXiv:2405.05941 (2024).
> [2] Chen, Siwei, et al. "Differentiable Particles for General-Purpose Deformable Object Manipulation." arXiv preprint arXiv:2405.01044 (2024).
>
> > Can you add one more baseline method of data augmentation to compare to?
>
> Thank you for the suggestion! Since we are proposing general augmentations that apply without imposing hard constraints on the tasks, we compare to previously proposed augmentation strategies  Laskin et al [1] and Chen et al [2]. In contrast to us, they use action invariant augmentations.
> Currently, given time constraints we managed to complete experiments with one of the tasks: close jar (see Table 1). In the meantime, we run more tasks. In close jar, PerAct obtained 37.2%, random patches on the RGB-D images as in [1] obtains an accuracy of 45.2%, randomly change the color of the table as in [2] lends 45.0%, finally inserting distractors as in [2] and [3] did not bring improvements with an accuracy of 36.4%. We recall that our approach Dream + Original obtains 51.2 in the close jar. We would include the results from the additional tasks we run, once they are ready.
> | Method                                         	| Accuracy (%) |
> |----------------------------------------------------|--------------|
> | PerAct                                                    	| 37.2  |
> | Random RGB-D patches [1]                	| 45.2  |
> | Random color [2]                                 	| 45.0  |
> | Distractors as in [2] and [3]                	| 36.4  |
> | Ours                                                       	| 51.2  |
>
>
>
>
> [1 ] Laskin, Misha, et al. "Reinforcement learning with augmented data." Advances in neural information processing systems 33 (2020): 19884-19895.
>
> [2] Chen, Zoey, et al. "Genaug: Retargeting behaviors to unseen situations via generative augmentation." arXiv preprint arXiv:2302.06671 (2023).
>
> [3] Marcel Torne, Anthony Simeonov, Zechu Li, April Chan, Tao Chen, Abhishek Gupta, and Pulkit Agrawal. Reconciling reality through simulation: A real-to-sim-to-real approach for robust manipulation. arXiv preprint arXiv:2403.03949, 2024

---

> ### Author Response · Authors · 2024-11-22
> **Part 2**
>
> > The method relies on open-vocabulary tracking models to segment objects, which limit the approach to non-articulated objects. It is unclear how such segmentation models can capture individual robot link or object parts connected with articulated joints accurately. Also, it is unclear how to extend the simulator to incorporate articulated objects interaction after the parts have been learned.
>
> > How to learn 3d models for parts of articulated objects and how to imagine demonstrations that manipulate articulated objects?
>
> The limitation raised by the reviewer is indeed valid. Extending the approach to articulated objects is an excellent direction that would require the agent to automatically predict three key aspects: the object's parts, the type of articulation, and the position of the articulation. While current open-vocabulary models are excellent at original object discovery, an additional split into parts could be grounded on object parts' motion [1] observed in demonstrations. In Appendix G of the updated paper, we provide a more detailed explanation and discuss potential approaches for extending the world model to handle articulated objects. We propose two possible approaches to address this challenge. The first involves using object semantics to directly predict articulations without requiring interactions. The second involves supervising the reconstruction process using the robot's trajectory and temporal data. We acknowledge this as an essential and interesting direction for future work.
> For the robot, as its URDF is available (as discussed in Section 3.3 ), we use segmented object parts from calibrated images, while the URDF model is provided by the manufacturer. Since the robot is calibrated with respect to the cameras, we can align the URDF to the learned Gaussians and thus group link Gaussians correspondingly. We apologize for the lack of clarity in our original explanation.
>
> [1] Unsupervised Discovery of Parts, Structure, and Dynamics, ICLR 2019
>
> > Existing approach of verify imagined demonstration is rudimentary.
>
> While the proposed verification is simple, it is effective in removing wrong demonstrations. We agree that for more challenging environments, more sophisticated approaches are interesting for future work [1].
>
> [1] Marius Memmel, Andrew Wagenmaker, Chuning Zhu, Dieter Fox, and Abhishek Gupta. Asid: Active exploration for system identification in robotic manipulation. In The Twelfth International Conference on Learning Representations, 2024
>
>
>
> ### Answers to questions
> > What’s the relationship with digital twin line of work?
>
> This is an interesting question. We believe that the concepts of world models and digital twins are converging. When a digital twin of a particular environment is automatically learned, it can effectively be considered a world model. By positioning our approach as a world model, we aim to foster greater awareness between the two communities—world models and real-to-sim/digital twins—and highlight their similarities. To address this connection, we discuss it in the related work section (Section 2.2) and provide further clarification and comparison in Appendix E.
>
> > Can you replay imagined trajectories in a simulator to verify correctness? Will that help improve imitation success?
>
> We replay the real trajectories in the world model to ensure that the final object positions in the augmented scenarios align with the expected outcomes ( the final position with the original trajectory transformed using corresponding equivariant transformation). This verification process is detailed in Section 4, where we discuss model validation. Moreover, the entry “Replay” in Table 2 indicates a PerAct agent trained only with DreMa replaying the original trajectory. Training a model with incorrect data often results in models that cannot perform tasks.
>
> >How does error build up in the pipeline of segmenting object masks -> learning objects models through Gaussian Spatting -> imagine demonstrations -> learning policy? Perhaps some ablations or quantitative metrics to measure error will be useful!
>
> A good segmentation is extremely important to obtain reliable data to train the agent. The prediction of the mesh is less crucial since a rough estimation could lead to a correct execution. However, wrong models could cause unexpected collisions. Finally, the model was checked using the original trajectory to obtain data with trajectories not executable by the robot arm. Even a small ratio of such data could highly impact the trained policy, obtaining a model not capable of learning the task. A deeper discussion is Appendix H.

---

> > ### Comment · Reviewer_kqVG · 2024-11-24
> > **Response**
> >
> > My concerns are mostly addressed by the rebuttal, so I am happy to raise the score.

---

> ### Author Response · Authors · 2024-11-28
> **Augmentations baselines**
>
> Dear Reviewer,
>
> Thank you for your thoughtful feedback and for recognizing the value of our work. We have conducted the requested experiments to compare PerAct with different types of invariant augmentations. The results are as follows:
>
> - PerAct achieved an average accuracy of 15.9% across all tasks without augmentations.
>
> - When trained with random patches on RGB-D images following [1], the accuracy increased to 17.8%.
>
> - Changing the table color [2] improved the average accuracy to 18.2%.
>
> - Introducing random distractors on the table [2,3] proved most beneficial, achieving an average accuracy of 20.1%, particularly improving the place cup and put groceries tasks compared to DreMa.
>
> While these augmentations improved performance, each approach had scenarios where the original PerAct outperformed them. By contrast, DreMa consistently surpassed PerAct, achieving an average accuracy of 25.1%, with a notable 5.0% improvement over PerAct's best invariant augmentation (distractors).
> Due to time constraints, we could not explore training PerAct with combinations of invariant and equivariant augmentations, which we believe could yield further improvements. This presents an exciting avenue for future work.
> We have incorporated these results into Section 5.1 and Table 1 and Appendix A of the manuscript. Thank you once again for your valuable suggestions, which have significantly enhanced our work. If you have any other questions we are happy to answer.
>
> [1] Laskin, Misha, et al. "Reinforcement learning with augmented data." Advances in neural information processing systems 33 (2020): 19884-19895.
>
> [2] Chen, Zoey, et al. "Genaug: Retargeting behaviors to unseen situations via generative augmentation." arXiv preprint arXiv:2302.06671 (2023).
>
> [3] Bharadhwaj, Homanga, et al. "Roboagent: Generalization and efficiency in robot manipulation via semantic augmentations and action chunking." 2024 IEEE International Conference on Robotics and Automation (ICRA). IEEE, 2024.

---

### Official Review · Reviewer_zmyw · 2024-11-04

**Soundness:** 2
**Presentation:** 2
**Contribution:** 2
**Rating:** 5
**Confidence:** 3

**Summary:**

This paper proposes DreMa, which integrates object-centric Gaussian splatting with a rigid-body physics simulator, to “imagine” new demonstrations to train imitation learning models. These imagined demonstrations are obtained in simulation by applying robot transformations and rotations to objects extracted from Gaussian splatting.

**Strengths:**

- The paper tackles the data efficient regime, and their proposed approach uses a single demonstration

- The paper validates the proposed approach with real-world experiments, and demonstrates a system that can perform manipulation tasks

**Weaknesses:**

My main issue with the paper is that the work is positioned within the “world model” literature, while the proposed method seems to fall under real2sim and data augmentation strategies. The “world model” is used to generate an augmented set of demonstrations in simulation in order to train an imitation learning model offline, and it is not used during online control. The set of equivariant transformations used to generate the augmented demo set is hand-designed, and likely task-specific.

The simulated results would be more convincing if they were expanded to include more than 3 RLBench tasks, but the method is limited to non-articulated objects. Additionally, the real-world tasks seem to only include blocks or boxes. How does DreMa perform with more complex shapes? How does DreMa perform when the scene includes a mix of objects with different shapes?

- It is unclear what base imitation learning model is used by the proposed DreMa method. Is this just PerAct? Are observations to the policy directly captured from cameras, or by rendering Gaussian splats?

- There are no metrics on runtime performance of the proposed method, while the introduction mentions the “real-time” performance of Gaussian splatting.

- Table 1: Comparisons to PerAct (Shridhar et al., 2023) are done using only 5 episodes per task, while PerAct uses 25 episodes per task.

- The related work section would benefit from a broader discussion of particle-based simulation approaches, such as: https://arxiv.org/abs/1810.01566, https://arxiv.org/abs/2312.05359, https://arxiv.org/abs/2405.01044, https://arxiv.org/abs/2303.05512. A comparison to ManiGaussian (https://arxiv.org/abs/2403.08321) would also be helpful.

**Questions:**

Figure 2: It would be helpful to include how “open-vocabulary tracking models” fit into the pipeline.

L107: “Two recent works demonstrated predicting future states can be applied to robotics” - This statement can be made more precise, since predicting future states / modeling forward dynamics for control is not a new idea in robotics.

L128: Why is (Cheng et al., 2023) used as a reference for the zero-shot capabilities of foundation models?

L157: There is some overloading of the term “manipulation”, which in the context of the paper seems to refer to a “manipulable” or controllable world model, rather than a world model explicitly designed for robot manipulation tasks.

L174: Set $\mathcal{X}$ notation is used for a sequence.

L267-268: Other objects in the world that the robot arm may interact with could also be articulated? ie. cabinets

L409: Is the PerAct baseline in the multi-task setting only trained on the subset of RLBench tasks you’re evaluating on? How many demonstrations were used? Was PerAct trained with data augmentation?

L412-413: Was this model selection using a validation set done over the entire course of training? How does this compare to just using the final model after training for 600k steps?

L418: Why “112, 61, and 81” demonstrations for the three tasks? How were these number of “imagined” demonstrations chosen?

L485: How were OOD test conditions chosen? Would they be within the set of equivariant transformations evaluated in Table 2?

[minor editing]

Figure 2: part 3, typo in Gaussian

L105: “gaussian” -> “Gaussian”

L233: $x_{n,k}$ should be $y$?

---

> ### Author Response · Authors · 2024-11-22
> **Part 1**
>
> Dear reviewer,
>
> Thank you for highlighting the paper's strengths, for your valuable suggestions, and, most importantly, for your constructive feedback, which helped to improve the paper significantly. We expanded the paper with more experiments to better demonstrate DreMa's contribution to the community, as well as clarifications in the main text and appendix. In addition, we address your questions directly here, referring to changes made in the paper.
>
> > My main issue with the paper is that the work is positioned within the “world model” literature, while the proposed method seems to fall under real2sim and data augmentation strategies.
>
> We understand how positioning our work within the “world model” literature might initially seem ambiguous, given its overlap with real2sim and data augmentation strategies. We believe our approach aligns with the definition of a world model as proposed by Ha and Schmidhuber [1], who describe it as a "spatial and temporal representation of the environment" capable of "predicting future sensory data." By positioning our model as a world model, we hope to make the two communities (world models and real2sim/digital twin one) more aware of each other and the similarities they might have. In addition, to cover this connection in the related work (Section 2.2), we clarify the connection and comparison between the two in the appendix (see App. E), showing how those areas are connected for robot manipulation tasks.
>
> Here, we summarize how DreMa satisfies key components of a world model:
>
> State: Represented as implicit latent vectors in traditional models, while in our case, it corresponds to explicit compositional representations that consist of the position of Gaussian splats, meshes, and other environment parameters.
>
> Observations: Traditionally inferred by a neural network (e.d. decoder), here derived from Gaussian Splatting renderings.
>
> Actions: Represented through neural network inputs or as end-effector positions, both applicable in our framework.
>
> > The “world model” is used to generate an augmented set of demonstrations in simulation in order to train an imitation learning model offline, and it is not used during online control.
>
> While the most apparent usage of the world model is for planning (which is a potentially interesting future application of DreMa), in this work, we show the main property of the world model ( "imagining or predicting realistic future sensory data from unseen states”) can be beneficial not only for planning but also for the offline training in the imitation learning, by exploiting world model predictions from unseen states.
>
> In conclusion, while our work incorporates elements of real2sim and data augmentation, we believe it fundamentally adheres to and extends the principles of world models. The growing overlap between these areas [2, 3] underscores the convergence of these methodologies for real-world robotics tasks, and we believe our contribution aligns with this evolution.
>
> [1] Ha, David, and Jürgen Schmidhuber. "World Models." arXiv preprint arXiv:1803.10122, 2018.
>
> [2] Abou-Chakra, Jad, et al. "Physically Embodied Gaussian Splatting: A Realtime Correctable World Model for Robotics." arXiv preprint arXiv:2406.10788, 2024.
>
> [3] Yang, Mengjiao, et al. "Learning Interactive Real-World Simulators." arXiv preprint arXiv:2310.06114, 2023.

---

> ### Author Response · Authors · 2024-11-22
> **Part 2**
>
> > The set of equivariant transformations used to generate the augmented demo set is hand-designed, and likely task-specific.
> > The simulated results would be more convincing if they were expanded to include more than 3 RLBench tasks.
>
> We understand the concern about the hand-designed nature of the transformations. Our approach, inspired by semantic segmentation practices in classical computer vision, uses augmentations like translations and rotations to improve generalization, which are also more often than not hand-designed rather than learnable. While hand-designed, the proposed augmentations are general and present valid ways to effectively enrich training distributions, which can be applied to many robot manipulation tasks.
> In robotics, particularly manipulation, applying augmentation to vision-based models is challenging, as noted by Pitis et al. [1]. Object manipulation affects both position and interaction actions, making data augmentation non-trivial due to task-specific needs and realistic RGB-D generation difficulties. Many approaches rely on manual modeling [2] or task-specific assumptions [3], emphasizing their specificity. Our method avoids these manual efforts by using Gaussian Splatting and decomposing the scene into objects, allowing the automatic generation of novel demonstrations and offering a scalable alternative for data augmentation in imitation learning.
> To further empirically demonstrate that those transformations are general, we expanded our evaluation to 9 RLBench tasks and 2 new real-robot tasks, while using the same set of transformations. DreMa significantly improves over baselines, 9% in RLBench and 29% with the real robot, showing its general applicability in generating useful training data for diverse manipulation tasks. Detailed results are in Table 1 and Table 4 of the updated manuscript.
>
> [1] Silviu Pitis et al. MoCoDA: Model-based counterfactual data augmentation. NeurIPS, 2022
>
> [2] Torne, Marcel, et al. "Reconciling reality through simulation: A real-to-sim-to-real approach for robust manipulation." arXiv preprint arXiv:2403.03949 (2024).
>
> [3] Mandlekar, Ajay, et al. "Mimicgen: A data generation system for scalable robot learning using human demonstrations." arXiv preprint arXiv:2310.17596 (2023).
>
>
> > Additionally, the real-world tasks seem to only include blocks or boxes. How does DreMa perform with more complex shapes? How does DreMa perform when the scene includes a mix of objects with different shapes?
>
> We note that the real-world experiments include a screwdriver and a star-shaped object, which are more complex than “blocks or boxes.”
>
> In the extra real robot tasks we reported, the first task we include is  “pick and place” of common objects with unusual shapes (i.e a tape, which is a hollow object, and a stapler, which is not exactly a box). We also include a second task where the robot needs to “erase” a colored spot. In these two new tasks, DreMa improves the baseline by 32,5% on average in in-domain and by 30% in out-of-domain settings. We include the detailed results in Table 4 in the updated manuscript. Appendix F visualizes and describes the tasks in more detail.
>
> > It is unclear what base imitation learning model is used by the proposed DreMa method. Is this just PerAct? Are observations to the policy directly captured from cameras, or by rendering Gaussian splats?
>
> We apologize for the unclear explanation in the paper. We use the name “DreMa” to refer to the world model and corresponding novel demonstration generation pipeline.
> As a base model for imitation learning, we use PerAct agent. With ‘DreMa’ in Table 1, we refer to a PerAct agent that is trained only on data from DreMa world model (thus using only the Gaussian splatting rendering data during training). With ‘DreMa + Original’, we refer to PerAct as trained with data generated by DreMa and the original demonstrations. We updated the paper to better differentiate this at the beginning of Section 5.1.
>
> > There are no metrics on the runtime performance of the proposed method, while the introduction mentions the “real-time” performance of Gaussian splatting.
>
> The runtime performance of our method is the same as PerAct agent, as we change only the training pipeline with novel data generated by the world model. In addition, in Appendix D, we report additional time needed for both the initial construction of the world model and the inference for novel states, comparing it with the inference using a standard PyBullet simulator without Gaussian Splatting rendering. We hope that this information will be useful for future work that may combine DreMa for different training regimes, such as RL fine-tuning.

---

> ### Author Response · Authors · 2024-11-22
> **Part 3**
>
> > Comparisons to PerAct (Shridhar et al., 2023) are done using only 5 episodes per task, while PerAct uses 25 episodes per task.
>
> First, we note that PerAct was trained in two regimes (10 and 100 demonstrations; see Sec 4.1 paragraph “Evaluation metric” of Shridhar et al., 2023), and was tested on 25 episodes ( Sec 4.1 paragraph “Evaluation metric” of Shridhar et al., 2023).
> In this work, we focus on a particularly challenging regime with a minimal number of demonstrations during training (5 episodes). However, we also show that our method brings significant benefits when more demonstrations (up to 20, including 10 used by the original PerAct) are available (see Figure 4). This clearly shows that our method is helpful in both settings, including the one on which PerAct was initially trained.
>
> > The related work section would benefit from a broader discussion of particle-based simulation approaches and a comparison to ManiGaussian
>
> We thank the reviewer for the suggestions. We added the suggested references in the updated manuscript, and we discussed differences in detail in L101.  In short, they can model more complex dynamics (Li et al., 2019), such as deformations, that are useful for robot manipulation Chen et al., 2024a). While previous approaches used neural radiance fields (Li et al., 2023; Whitney et al., 2024), current approaches exploit the explicit representation of gaussian splatting (Xie et al., 2024; Jiang et al., 2024).  In L148, we discuss how ManiGaussian proposes an agent powered by Gaussian Splatting, instead, we propose a world model capable of generating novel training data.
>
>
> ### Answers to questions
>
> > Figure 2: It would be helpful to include how “open-vocabulary tracking models” fit into the pipeline.
>
> We updated Figure 2, highlighting that the segmentation is done on the sequence of images to better represent the tracking contribution (i.e., predicting consistent masks for the whole sequence of images).
>
> >...This statement can be made more precise, since predicting future states / modeling forward dynamics for control is not a new idea in robotics.
>
> Thank you for catching this. We changed to “two recent studies showed that scene reconstruction can enhance robot policies~\citep{ruan2024primp, torne2024reconciling}.” to be more specific.”

---

> ### Author Response · Authors · 2024-11-22
> **Part 4**
>
> > L128: Why is (Cheng et al., 2023) used as a reference for the zero-shot capabilities of foundation models?
>
> Cheng et al, 2023 showed how to combine several foundational models (e.g. SAM + GroundingDINO) and apply them for temporal data to get consistent zero-shot training of the object (new ability to track objects consistently). In the revised version, we add in L134 the missing references (e.g. Kirillov et al., 2023) to the original foundational models.
>
> > L412-413: Was this model selection using a validation set done over the entire course of training? How does this compare to just using the final model after training for 600k steps?
>
> We follow the same procedure as PerAct (Appendix C.1 of Shridhar et al., 2023). We train on the training set and validate on an entirely different validation set for model selection.
> If we use the last weights, we often overfit to the training data, like with any other deep learning downstream task. For example, in the slide block task (single task), the last model of PerAct obtains 37.5% accuracy in the validation set. The selected model obtains 52.5% accuracy instead. In the same task, Drema + Original obtained 62.5% accuracy and 67.5%, respectively.
>
> > L418: Why “112, 61, and 81” demonstrations for the three tasks? How were these number of “imagined” demonstrations chosen?
>
> To generate data with DreMa, we transform each of the original demonstrations to all possible transformations (i.e. 8 translations, 12 rotations, 18 object rotations see App. L paragraph Data Generation of updated paper). Next, we execute the augmented trajectory and compare the object's final position in both transformed and original environments. The augmented demonstration is kept if it is valid (i.e the final positions are similar). This process is now described better in L417.
>
> > There are no metrics on runtime performance of the proposed method, while the introduction mentions the “real-time” performance of Gaussian splatting.
>
> The proposed method has two parts: agent trained with world model simulation and world model itself (DreMa); below, we cover inference and training time for both of them.
> PerAct training and inference. Training: PerAct takes two days on an Nvidia A40 for 100k iterations, making real-time performance impractical. Inference: PerAct inference is the same as that of the original method (~2 frames per second).
> DreMa training and inference. Training: DEVA-tracking processes five images per second while generating Gaussian models and meshes scales linearly with object count, taking about four minutes per object. Inference: Simulations are faster, averaging 1.715 seconds to reach a waypoint and 0.168 seconds to render five 128×128 RGB-D images.
> Detailed comparisons are provided in Appendix D.
>
> > There is some overloading of the term “manipulation”, which in the context of the paper seems to refer to a “manipulable” or controllable world model, rather than a world model explicitly designed for robot manipulation tasks.
>
> We will reconsider the terminology when preparing the camera-ready version, and carefully consider to update the term manipulable with controllable to improve the clarity as suggested.
>
> >​​ L485: How were OOD test conditions chosen? Would they be within the set of equivariant transformations evaluated in Table 2?
>
> In the task description of App F.2, we explain that for the OOD test. We imposed some structure constraints in the collected data (for example the cube is always between the targets in the pick block task). In the OOD we randomize the initial and target positions of objects. As their positions are random, they are not constrained to be in the transformed positions.
>
> > is the PerAct baseline in the multi-task setting only trained on the subset of RLBench tasks you’re evaluating on? How many demonstrations were used? Was PerAct trained with data augmentation?
>
> In the multi-task setting, PerAct is trained simultaneously on multiple tasks, enabling it to handle a broader range of tasks. We followed the original data augmentation method proposed by PerAct’s authors, including random translations of up to 0.125 meters and random rotations along the z-axis up to 45 degrees. The number of original training demonstrations is 5 for each task except for slide block and place wine that have a reduced number of variations (respectively 4 and 3), resulting in 42 demonstrations in total  (see L402).

---

> > ### Comment · Reviewer_zmyw · 2024-11-24
> > **Evaluation methodology and framing the approach**
> >
> > > We believe our approach aligns with the definition of a world model as proposed by Ha and Schmidhuber [1], who describe it as a "spatial and temporal representation of the environment"
> >
> > > Our method avoids these manual efforts by using Gaussian Splatting and decomposing the scene into objects, allowing the automatic generation of novel demonstrations and offering a scalable alternative for data augmentation in imitation learning.
> >
> > Generally within world model literature, like in [Ha and Schmidhuber, 2018], the world model is trained to "*learn* a compressed spatial and temporal representation of the environment". Based on my understanding the paper, the proposed method uses Gaussian splatting to extract a mesh for use in a physics simulation, and to render observations from the augmented set of demonstrations. In this case, "imagined demonstrations" are hand-designed and dependent on what is within the pre-defined set of augmentations.
> >
> > I think that either replacing dynamics with a learned model instead of physics-based simulation, or predicting a set of transformations based on the task (instead of using a hand-designed set) that are used to produce "imagined" demonstrations, would more accurately fall within the "world modeling" domain. Given the current description of DreMa, it is more accurate to strictly call it a real2sim and data augmentation strategy (and I think that would be ok! I think readers would appreciate that presentation more than claiming DreMa is a world modeling approach).
> >
> > > In the extra real robot tasks we reported, the first task we include is “pick and place” of common objects with unusual shapes (i.e a tape, which is a hollow object, and a stapler, which is not exactly a box).
> >
> > Given the low resolution of the images in Figure F.9, it is very difficult to determine what these tasks are and whether they are successfully performed.
> >
> > For "place object", it appears like the tape dispenser is still being held by the gripper in the final 5th image. For "push block" with the screwdriver, how does the agent re-orient the screwdriver between the 2nd and 3rd images?
> >
> > > Comparisons to PerAct (Shridhar et al., 2023) are done using only 5 episodes per task, while PerAct uses 25 episodes per task.
> >
> > > ... In this work, we focus on a particularly challenging regime with a minimal number of demonstrations during training (5 episodes). ...
> >
> > My point is that PerAct uses 25 evaluation episodes per task, while your main results (Table 1, Table 2) are reported using only 5 evaluation episodes / test runs per task.
> >
> > Aside: There is still a typo in Figure 2, "Physic-powered" -> "Physics-powered" or Physics-based

---

> > > ### Author Response · Authors · 2024-11-24
> > > **Response to: Evaluation methodology and framing the approach (Part 1/2)**
> > >
> > > > “Generally within world model literature, like in [Ha and Schmidhuber, 2018], the world model is trained to "learn a compressed spatial and temporal representation of the environment". Based on my understanding the paper, the proposed method uses Gaussian splatting to extract a mesh for use in a physics simulation, and to render observations from the augmented set of demonstrations. In this case, "imagined demonstrations" are hand-designed and dependent on what is within the pre-defined set of augmentations.
> > >
> > > > I think that either replacing dynamics with a learned model instead of physics-based simulation, or predicting a set of transformations based on the task (instead of using a hand-designed set) that are used to produce "imagined" demonstrations, would more accurately fall within the "world modeling" domain. Given the current description of DreMa, it is more accurate to strictly call it a real2sim and data augmentation strategy (and I think that would be ok! I think readers would appreciate that presentation more than claiming DreMa is a world modeling approach).”
> > >
> > > We acknowledge your point. From what we understand, in your view we must have clear end-to-end learning, either in the dynamics (e.g. replace the physics simulator) or in the ‘generation’ of novel world states. So, it appears (to us) that we mainly disagree on the positioning and on the semantics of what is or should be a world model.
> > > In our view, our model primarily relies on lots of core “spatial representation learning” components, many of which are learned end-to-end, and that we use out of the box and combine to learn such spatial representations.
> > >
> > > Regarding the scene and object representations:
> > >
> > > - The scene decomposition relies heavily on learned components, including SAM, an open-vocabulary GroundingDINO, and DEVA. These components are critical to ensuring consistency and adaptability in the scene representation.
> > >  - Most importantly, the mesh extraction process involves reconstruction from RGB-D sensory inputs, which itself is a learned process derived directly from observations of the environment.
> > >
> > > Regarding the dynamics, they are indeed a "non-learnable" component that relies heavily on the inductive bias of the hand engineered simulator. However, we do not view this as conflicting with the definition of a world model for the following reasons:
> > >
> > > - Known and Fixed Newtonian Mechanics: In a robot’s environment, dynamics are governed by physical laws. Neural networks excel at approximating abstract rules from observed data, but when the underlying rules (e.g., Newtonian mechanics) are known and invariant, explicitly incorporating these into the model (via simulators) is a principled choice. This approach ensures better generalization and reduces the risk of hallucinations or inaccuracies.
> > >
> > > - Task-Driven Learning: While transformations are predefined, our approach filters these transformations through task constraints, automatically discarding trajectories that lead to invalid outcomes. This filtering is a form of learning, as it selects and refines data used for policy learning based on the agent’s objectives. We agree that introducing more sophisticated search or gradient-based optimization could enhance accuracy and robustness. We view this as an exciting avenue for future work and hope it inspires further exploration by the community.
> > >
> > > Last, we are not dogmatic in our choice of terminology. While we believe that framing DreMa as an “explicit world model” highlights its unique design choices and contributions, we respect your perspective. Since the other two reviewers did not focus on this as much, if the above explanations were still not convincing and the AC agrees with your view, we are open to rephrasing and positioning DreMa as a “real2sim” approach. We hope that we clarified all the points.
> > >
> > > Thank you again for your constructive critique. We acknowledge and value your feedback and look forward to further engaging with the community on this topic.

---

> > > > ### Author Response · Authors · 2024-11-24
> > > > **Response to: Evaluation methodology and framing the approach (Part 2/2)**
> > > >
> > > > > Given the low resolution of the images in Figure F.9, it is very difficult to determine what these tasks are and whether they are successfully performed. For "place object", it appears like the tape dispenser is still being held by the gripper in the final 5th image. For "push block" with the screwdriver, how does the agent re-orient the screwdriver between the 2nd and 3rd images?
> > > >
> > > > Apologies. We will include high-resolution images tomorrow in the appendix. We also added to the website (https://dreamtomanipulate.github.io/DreMa/) the new videos of the robot performing all the requested tasks (including more complex shape tasks), where the process of the screwdriver reorientation is fully visible.
> > > >
> > > > > My point is that PerAct uses 25 evaluation episodes per task, while your main results (Table 1, Table 2) are reported using only 5 evaluation episodes / test runs per task.
> > > >
> > > > Thank you for pointing out the unclear explanation in the paper. To clarify, we use 50 test examples per task, an increase from the 25 used in PerAct. This choice was made to better capture the diverse variations in the tasks, such as the 60 possible variations for the stack block task, and report more reliable numbers.The 5 iterations mentioned in our paper refers to the number of times the testing process was repeated per each task. That is, per task, we run the model for 50x5=250 times to report the final numbers. These iterations are to account for variability introduced by the motion planner and the interaction with the environment. We have included this information in L405. We also clarify the descriptions in Table 1 and 2. We also explained this in Appendix L.
> > > >
> > > > > Aside: There is still a typo in Figure 2, "Physic-powered" -> "Physics-powered" or Physics-based
> > > >
> > > > Thank you again for revising the paper and finding the typo. We updated the figure.

---

> > > ### Author Response · Authors · 2024-11-28
> > >
> > > Dear Reviewer,
> > >
> > > We noticed that we have not received any follow-up to our previous responses. If you have any further questions or require additional clarifications, we would be pleased to address them. Thank you for your time and consideration.

---

> ### Author Response · Authors · 2024-11-25
> **Changed resolution**
>
> Dear reviewer,
>
> we updated the images of the paper increasing the resolution. Thank you for your feedback.

---

> ### Author Response · Authors · 2024-12-02
>
> Dear Reviewer,
>
> As the extended discussion period approaches its conclusion, we would greatly appreciate receiving your feedback at your earliest convenience. If our rebuttal has addressed your concerns satisfactorily, we kindly request an update on your evaluation. Your further input is valuable, and we remain available to address any additional questions or concerns you may have.
>
> Thank you for your time and consideration.
>
> Sincerely,
>
> the Authors

---

> > ### Comment · Reviewer_zmyw · 2024-12-02
> >
> > I appreciate the time and effort that the authors took in preparing the rebuttal. I will maintain my current rating, given that my overall concern on clarity and presentation are still remaining. I disagree with the presentation of the proposed approach as a "compositional manipulation world model" (Aside: "manipulation" world model is still overloaded in the revised manuscript). I would recommend the authors more carefully consider how their work is situated against existing literature on real2sim, data augmentation, domain randomization, and learned world models. After this discussion and further consideration, the proposed approach still seems to fall under explicit real2sim and data augmentation strategies. The physics simulation parameters are pre-set, the set of transformations is chosen for the tabletop manipulation tasks being considered, and an additional filtering step is used on top to obtain the final augmented set of demonstrations; whereas GS is used for inverse rendering for system identification and rendering given an explicit physics simulation. This is done per-scene / per-demonstration.

---

> > > ### Author Response · Authors · 2024-12-04
> > >
> > > Dear Reviewer,
> > >
> > > Thank you for your thoughtful feedback and concerns.
> > >
> > > - The need to collect new observations when transitioning between environments is common in world models, including DreamerV2 [1], which requires re-optimizing its network when switching between Atari games. Similarly, our approach gathers observations and re-optimizes when encountering new tasks and  environments.
> > >
> > > - Regarding your suggestion to position our work under real-to-sim (real2sim) or data augmentation frameworks:
> > > Data Augmentation:
> > >
> > >   - Traditional augmentation modifies existing data (e.g., patches or color changes). In contrast, our method generates entirely new data based on DreMa’s state predictions, creating novel trajectories using learned representations. This is conceptually distinct from standard augmentation, as it involves imagining new possibilities rather than altering existing ones.
> > >
> > >   - Real2Sim: Real2Sim aims to create high-fidelity simulations, often requiring manual intervention [2] or not using the reconstructed environment actively [3]. Our approach differs by leveraging state predictions from a world model to facilitate task learning, focusing on autonomous generation of novel demonstrations rather than recreating precise environmental replicas. While world models may seem real2sim-like (e.g., DreamerV2 reconstructing aspects of Atari games), they autonomously learn from observational data to enable agent learning. Applications like DayDreamer [4] blur this distinction further when used with real robots.
> > >
> > > - Hand-Designed Components: Our approach incorporates predefined components like most world models, (e.g., reward functions and inductive biases in neural architectures). However, DreMa autonomously generates data for new tasks without manual adjustments, consistently improving baseline performance (+9.1%) and outperforming augmented baselines (+5%).
> > >
> > > - While our parameters are fixed, Appendix J discusses how they could be updated using related methods. The goal of DreMa is not to perfectly replicate the real environment but to enable the agent to effectively learn directly from DreMa’s predictions.
> > >
> > >
> > > Thank you for your constructive insights, and we hope this clarifies our perspective.
> > > The Authors
> > >
> > > [1] Hafner, Danijar, et al. "Mastering Atari with Discrete World Models." International Conference on Learning Representations 2021.
> > >
> > > [2] Marcel Torne, Anthony Simeonov, Zechu Li, April Chan, Tao Chen, Abhishek Gupta, and Pulkit Agrawal. Reconciling reality through simulation: A real-to-sim-to-real approach for robust manipulation. arXiv preprint arXiv:2403.03949, 2024
> > >
> > > [3] Allan Zhou, Moo Jin Kim, Lirui Wang, Pete Florence, and Chelsea Finn. NeRF in the palm of your hand: Corrective augmentation for robotics via novel-view synthesis. CVPR 2023
> > >
> > > [4] Wu, Philipp, et al. "Daydreamer: World models for physical robot learning." Conference on robot learning. PMLR, 2023.

---

### Author Response · Authors · 2024-12-04
**General response and summary of rebuttal**

Dear Reviewers, Dear AC,

Thank you for all your help to bring this paper forward. We believe it introduces a new way to ground world representations with the physical world, and trying to fit into a single and specific category is not the point. We think this is evident from the substantial improvements (up to 50% absolute improvements in some tasks, 33.3% on average) with few-shot imitation learning with real robots, not only on the original 3 tasks, but also 2 more that we run during the rebuttal.

Reviewers kqVG and Mne8 acknowledge our message and positioning (Mne8: “Congratulations, I think your work is valuable.”).

As far as we can tell, the only remaining difference in perspective
is with Reviewer zmyw on the positioning: is the proposed model a world model (as we claim), a data augmentation model or a real2sim model. We summarize why we think our proposed model should be better thought of as a world model, and why a ‘data augmentation’ or ‘real2sim’ positioning does not perfectly fit.

**Why call it a world model approach?**

- A world model is a learned representation, and our work includes learning components.

-  Future state predictions, observations and actions are key characteristics present in our approach.

- We demonstrated that the agent could learn directly within the learned its “own hallucination" [5].
- Our approach offers benefits beyond planning, including offline training in imitation learning, by leveraging world model predictions from unseen states.

- Prior works such as [6] and [7] support using world model definitions in our context.

 - As we discussed in Sections 2.1 and 2.2, along with Appendix E, there is a strong connection between world models in robotics and real2sim approaches.

- Existing world models also include hand-designed components, such as reward functions and inductive biases in neural architectures.

- DreMa autonomously generates imagined data for new tasks without manual adjustments, consistently improving baseline performance (+9.1%) and outperforming the traditional augmented baselines (+5%).

- The need to collect new observations when transitioning between environments is also common in world models, including DreamerV2 [1], which requires re-optimizing its network when switching between Atari games. Similarly, our approach gathers observations and re-optimizes when encountering new tasks and  environments.

**Why not call it only a data augmentation approach?**

- Traditional augmentation modifies existing data (e.g., patches or color changes). In contrast, our method generates entirely new data based on DreMa’s state predictions, creating novel trajectories using learned representations.

- The DreMa process is conceptually distinct from standard augmentation, as it involves imagining new possibilities rather than altering existing ones.

**Why not call it only a real2sim approach?**

- Real2Sim aims to create high-fidelity simulations, often requiring manual intervention [2] or not using the reconstructed environment actively [3]. Our approach differs by leveraging state predictions from DreMa to facilitate task learning, focusing on autonomous generation of novel demonstrations rather than recreating precise environmental replicas.

- While world models may seem real2sim-like (e.g., DreamerV2 reconstructing aspects of Atari games), they autonomously learn from observational data to enable agent learning. Applications like DayDreamer [4] blur this distinction further when used with real robots.

In the end, we think that an added value comes from comparing different sub-fields, finding what is common between them rather than what is not, and bridging communities.

Thank you for your constructive insights, and we hope this clarifies our perspective.

The Authors

[1] Hafner, Danijar, et al. "Mastering Atari with Discrete World Models." International Conference on Learning Representations 2021.

[2] Marcel Torne, Anthony Simeonov, Zechu Li, April Chan, Tao Chen, Abhishek Gupta, and Pulkit Agrawal. Reconciling reality through simulation: A real-to-sim-to-real approach for robust manipulation. arXiv preprint arXiv:2403.03949, 2024

[3] Allan Zhou, Moo Jin Kim, Lirui Wang, Pete Florence, and Chelsea Finn. NeRF in the palm of your hand: Corrective augmentation for robotics via novel-view synthesis. CVPR 2023

[4] Wu, Philipp, et al. "Daydreamer: World models for physical robot learning." Conference on robot learning. PMLR, 2023.

[5] Ha, David, and Jürgen Schmidhuber. "World Models." arXiv preprint arXiv:1803.10122, 2018.

[6] Yang, Mengjiao, et al. "Learning Interactive Real-World Simulators." The Twelfth International Conference on Learning Representations, 2024.

[7] Abou-Chakra, Jad, et al. "Physically Embodied Gaussian Splatting: A Realtime Correctable World Model for Robotics." arXiv preprint arXiv:2406.10788, 2024.

---

### Meta-Review · Area_Chair_ty9b · 2024-12-19

**Metareview:**

This paper introduces a novel method for constructing photo-realistic 3D simulations (world models) of real-world scenes using object-centric Gaussian splatting. The resulting simulation is then used to generate data for augmenting few-shot imitation learning.

The reviewers agree that the method is novel and impactful. However, key concerns raised include: (1) the positioning of the paper as a world-model approach versus a real2sim data augmentation method, (2) insufficient comparisons with relevant baselines, and (3) limited evaluation on simple pick-and-place tasks. Concerns regarding the paper's clarity were addressed during the discussion phase.

Regarding point (1), I find that the current presentation effectively communicates the method's intent and does not risk confusing the audience. Nonetheless, I concur with reviewer zmywwe's suggestion to downplay the emphasis on the world-model aspect to avoid potential misalignment with the paper's contributions.

On point (2), the evaluation against baselines and design choice ablations appear insufficient for the following reasons:
- The choice of PerAct as the downstream policy learning algorithm does not fully capitalize on the photo-realistic renderings generated by Gaussian splatting, raising questions about whether this choice aligns with the method's strengths.
- Additionally, the same data augmentation approach could potentially be applied to segmented voxels or point clouds, suggesting that the benefits of Gaussian splatting for this task remain underexplored in comparison.

Point (3) is a major limitation, the authors should further clarify and elaborate how they envision the proposed approach may generalize to articulated objects.

Overall, I believe addressing these gaps in evaluation and contextualization would strengthen this paper further.

**Additional Comments On Reviewer Discussion:**

The authors clarified questions raised during initial review period and one reviewer raised their score during discussion.

---

### Decision · Program_Chairs · 2025-01-22

Accept (Poster)